# Cloud droplet diffusional growth in homogeneous isotropic turbulence: bin microphysics versus Lagrangian superdroplet simulations

Wojciech W. Grabowski[1] and Lois Thomas[2,3]

[1]Mesoscale and Microscale Meteorology Laboratory, National Center for Atmospheric Research,
 Boulder, CO 80307, USA
[2]HPCS, Indian Institute of Tropical Meteorology, Ministry of Earth Sciences, Pune 411008, India
[3]Department of Atmospheric and Space Sciences, Savitribai Phule Pune University, Pune 411007, India

*Correspondence to*:  W. W. Grabowski (grabow@ucar.edu)

**Abstract.** Increase of the spectral width of initially monodisperse population of cloud droplets in homogeneous isotropic turbulence is investigated by applying a finite-difference fluid flow model combined with either Eulerian bin microphysics or Lagrangian particle-based scheme.

The turbulence is forced applying a variant of the so-called linear forcing method that maintains the mean turbulent kinetic energy (TKE) and the TKE partitioning between velocity components. The latter is important for maintaining the quasi-steady forcing of the supersaturation fluctuations that drive the increase of the spectral width. We apply a large computational domain, $64^3$ m$^3$, one of the domains considered in Thomas et al. (2020). The simulations apply 1 m grid

length and are in the spirit of the implicit large eddy simulation (ILES), that is, with small-scale dissipation provided by the model numerics. This is in contrast to the scaled-up direct numerical simulation (DNS) applied in Thomas et al. (2020). Two TKE intensities and three different droplet concentrations are considered. Analytic solutions derived in Sardina et al. (2015), valid for the case when the turbulence integral time scale is much larger than the droplet phase

relaxation time scale, are used to guide the comparison between the two microphysics simulation techniques. The Lagrangian approach reproduces the scalings relatively well. Representing the spectral width increase in time is more challenging for the bin microphysics because appropriately high resolution in the bin space is needed. The bin width of 0.5 μm is only sufficient for the lowest droplet concentration, 26 cm$^{-3}$. For the highest droplet concentration,

650 cm$^{-3}$, an order of magnitude smaller bin size is barely sufficient. The scalings are not expected to be valid for the lowest droplet concentration and the high TKE case, and the two

microphysics schemes represent similar departures. Finally, because the fluid flow is the same for all simulations featuring either low or high TKE, one can compare point-by-point simulation results. Such a comparison shows very close temperature and water vapor point-by-point values across the computational domain, and larger differences between simulated mean droplet radii and spectral width. The latter are explained by fundamental differences in the two simulation methodologies, numerical diffusion in the Eulerian bin approach and relatively small number of Lagrangian particles that are used in the particle-based microphysics.

## 1 Introduction

Cloud droplet spectra in natural ice-free clouds significantly affect such key processes as drizzle/rain formation and transfer of solar radiation through the cloudy atmosphere. At the same time, modeling of droplet spectra is cumbersome and thus simplified approaches are often used, such as the bulk microphysics where the shape of the droplet spectrum is prescribed or not considered at all. When simulation of the droplet spectral shape is required, there are two basic modeling methodologies that can be used. The first one is a traditional bin approach where the Eulerian continuous in space and time spectral density function is used. In its numerical implementation, the spectral density function is represented by a finite number of radius (or mass) bins. Each bin is advected in the physical space and all bins are combined at model grid locations to calculate the change of the spectral density function due to droplet growth. The bin microphysics is a well-established approach to modeling droplet spectral evolution, see Khain et al. (2015) and references therein. The second approach represents the multiphase nature of real clouds by applying Lagrangian point particles. Each particle represents an ensemble of natural droplets with the same properties, it is advected by the simulated air flow, and it grows in response to local conditions. The Lagrangian approach, often referred to as the super-droplet method (Shima et al., 2009), is a relatively novel modeling technique that gains popularity in cloud modeling because of its fidelity, especially for the simulation of aerosol-cloud interactions (e.g., Andrejczuk et al., 2008; Shima et al., 2009; Riechelmann et al., 2012; Arabas and Shima, 2013; Unterstrasser et al., 2017; Hoffmann et al., 2019; Dziekan et al., 2019; see also Grabowski

et al., 2019). The Lagrangian approach is referred to as the "swarm model" in the astrophysical context (see Li et al., 2017 and references therein).

The two methodologies have their inherent limitations. The bin microphysics is affected by the numerical diffusion as any Eulerian approach. Advection of bins in the physical space typically leads to unavoidable numerical spreading of regions with rapid droplet spectral changes, for instance, near cloud edges. The diffusional growth of cloud droplets is represented by the advection of the spectral density function in the radius (or mass) space and it is impacted by

numerical aspects similar to the advection in the physical space (e.g., section 3.1 in Li et al., 2017). The combined effect of the advection in the physical space and advection in the radius space is argued by Morrison et al. (2018) to result in artificial broadening of the droplet spectra in cloud simulations applying bin microphysics. For the Lagrangian microphysics, an obvious limitation is the limited and usually small number of Lagrangian particles that can be afforded in

realistic cloud simulations, especially considering an enormous number of cloud and precipitation particles in natural clouds. However, the Lagrangian methodology has clear benefits when compared to the bin scheme. These include the lack of numerical diffusion, realistic representation of the stochastic nature of the cloud droplet growth, possibility of including physically-based representation of the unresolved scales impact on droplet growth (i.e.,

allowing the multiscale simulation of a turbulent cloud), and providing a better framework for aerosol-cloud interactions and representation of ice processes. Grabowski et al. (2019) provide a review of these benefits.

    Grabowski (2020a, 2020b; G20a and G20b, respectively) compared cloud droplet activation and

growth by the diffusion of water vapor in simulations of a laboratory cloud chamber and a single cumulus congestus cloud, respectively. The laboratory cloud chamber at the Michigan Technological University (see http://phy.sites.mtu.edu/cloudchamber/) forms a cloud because of the temperature and humidity differences between lower and upper horizontal boundaries that drive turbulent Rayleigh–Bénard convection. G20a shows a good agreement between droplet

spectra predicted by the two methodologies when averaged over the chamber volume away from boundaries. G20a argued that the good agreement was because of the constant chamber pressure assumed in the simulations. This agrees with Morrison et al. (2018) conjecture that the artificial

spectral broadening comes from the coupling between vertical advection in a stratified environment (that provides the supersaturation source) and advection in the bin space that represents response of the droplet population to the supersaturation forcing. Cumulus congestus case from G20b is based on a modeling study by Lasher-Trapp et al. (2005) that considered a cloud observed by a radar and an instrumented aircraft during the Small Cumulus Microphysics Study (SCMS) near Cape Canaveral, Florida, during July–August of 1995. A unique aspect of the G20b study is application of the piggybacking methodology. Piggybacking refers to using two microphysics schemes in a single cloud simulation, one scheme driving the dynamics and the other one piggybacking the simulated flow, see Grabowski (2019) for a review. Operating the two schemes in the same cloud-scale flow allows point-by-point comparison of droplet spectra predicted by the two schemes. A significantly larger mean spectral width simulated by the bin scheme across the entire cloud depth is the largest difference between the two schemes in G20b simulations.

In this paper, we discuss differences between the two methodologies for representing droplet spectral evolution in numerical homogeneous isotropic turbulence. Li et al. (2017) present similar comparisons applying dynamic and kinematic simulations, and including collision-coalescence. Here, we consider only diffusional growth of cloud droplets. The direct numerical simulation (DNS) methodology (e.g., Vaillancourt et al., 2001, 2002; Lanotte et al., 2009; Li et al., 2019) and scaled-up DNS technique (Thomas et al., 2020) allow representation of turbulence impact on the droplet spectral width with an unprecedented fidelity. Sardina et al. (2015) and Grabowski and Abade (2017) provide stochastic model reference for such studies (cf. Fig. 10 in Thomas et al., 2020). In contrast to Thomas et al. (2020) who used a traditional spectral DNS code, we apply a finite-difference fluid flow model that does not require small-scale dissipation to maintain computational stability. It follows that the simulations are in the spirit of the implicit large eddy simulation (ILES) where the model numerics provide the required small-scale dissipation of the turbulent kinetic energy (TKE) and scalar variance. Details of the fluid flow model are presented in the next section with the emphasis on the forcing to maintain the quasi-steady turbulence, the key element of the homogeneous isotropic turbulence DNS. Two turbulence cases are considered, the low TKE case (following Lanotte et al., 2009 and Thomas et al., 2020) and the high TKE case, the latter featuring hundred times larger TKE than the former.

Section 3 introduces the temperature and water vapor equations for moist simulations and

presents results from simulations without droplets. Section 4 introduces numerical representation
of cloud droplets applying either Eulerian bin microphysics or Lagrangian superdroplets. Results
of simulations with droplets are presented in section 5 focusing on the ability of either scheme to
represent theoretical scalings derived by Sardina et al. (2015) and on the comparison of the
droplet spectra simulated by the two schemes. Section 6 shows grid-volume by grid-volume

comparison of model results facilitated by the simulation methodology, exposing additional
limitations of the two microphysics simulation approaches. A brief summary in section 7
concludes the paper.

**2 Homogeneous isotropic turbulence simulations**


**2.1 The model and model forcing**

The EULerian--semi-LAGrangian (EULAG) anelastic finite-difference fluid flow model
([http://www.mmm.ucar.edu/eulag/](http://www.mmm.ucar.edu/eulag/)) is used in this study in the ILES mode (Margolin and Rider,

2002; Andrejczuk et al., 2004; Margolin et al., 2006; Grinstein et al., 2007). ILES implies that
the model uses no explicit dissipation and removes small-scale velocity and scalar fluctuations
through numerical diffusion provided by the monotone advection scheme. The fluid flow
equations for homogeneous isotropic turbulence simulations are (e.g., Lanotte et al., 2009, Li et
al., 2017):


$$\frac{\partial u}{\partial t} + \text{div } (\boldsymbol{u}.\boldsymbol{u}) = \text{-}1/\rho \text{ grad } p + \boldsymbol{f} \qquad (1)$$

$$\text{div } \boldsymbol{u} = 0, \qquad\qquad (2)$$

where $\boldsymbol{u}$ is the fluid flow velocity, $p$ is pressure, $\rho = 1$ kg m$^{-3}$ is the air density, and $\boldsymbol{f}$ is the
turbulence forcing term. The forcing term ensures that the turbulence is maintained throughout
the simulation with TKE flowing from large-scales towards the small-scale dissipation. The
traditional technique to force the quasi-equilibrium homogeneous isotropic turbulence,
convenient for spectral models, is to consider the forcing only for a few low-wavenumber modes.

However, such an approach is not practical for the finite-difference model used here. Instead, we apply a method in the spirit of the so-called linear forcing of Rosales and Meneveau (2005) and Onishi et al. (2011). In the homogeneous isotropic turbulence, TKE increases with the eddy size $L$ as $L^{2/3}$, that is, TKE is dominated by contributions from the largest eddies. Hence, one can force the turbulence by simply ensuring that the TKE does not change from one model time step

to the next one because such forcing affects mostly large eddies. This implies that $u^{(n+1)} = \alpha \, u^{(n)}$ with $\alpha = (E_t/E^{(n)})^{1/2}$, where $n$ and $n+1$ represent time levels, $E^{(n)}$ is TKE at the $n$ time level, and $E_t$ is the target TKE (see Eq. 3 in Onishi et al., 2011). The finite difference representation of such a forcing is given by

$$f = (u^{(n+1)} - u^{(n)})/\Delta t = u^{(n)}(\alpha - 1) \, / \, \Delta t \ , \quad (3)$$

that is, as in the case of the linear forcing of Rosales and Meneveau (2005). The TKE dissipation rate $\varepsilon$ can be derived assuming that the TKE does not change in time [see (5) in Onishi et al., (2011)] as


$$\varepsilon = 2 \, E_t (\alpha - 1) \, / \, \Delta t \ . \quad (4).$$

Eq. 4 is particularly useful in ILES because it allows diagnosing the TKE dissipation rate that is otherwise not known.


The forcing described above was initially applied in dry ILES simulations, that is, the EULAG model solving (1) and (2). For those tests (and for other simulations described in this paper), the initial flow field (scaled-up to approximately match the required TKE) was taken as the initial flow pattern in decaying turbulence simulations in Andrejczuk et al. (2004), see Fig. 1 therein.

Other parameters of the test simulations correspond to low TKE setup as described in the next section. Those initial tests forced as in (3) revealed the need for additional forcing modifications as discussed below.

As in other studies of forced homogeneous isotropic turbulence (e.g., Lanotte et al., 2009), the

model applies computational domain with triply-periodic lateral boundary conditions. Such

boundary conditions together with (2) imply that the mean flow across the domain has to be uniform. For instance, $d\langle u_z \rangle_{xy}/dz$ (where $u_z$ is the vertical velocity and $\langle . \rangle_{xy}$ is the horizontal average) has to vanish, and the same is true for the other two spatial directions. However, the vertically-uniform $\langle u_z \rangle_{xy}$ can evolve in time. In the initial tests, a gradual development of the

mean flow across the domain was noticed. In other words, in addition to driving the turbulence inside the computational domain, the forcing (3) resulted in a gradual development of the mean flow across the domain. To eliminate this undesirable behavior, an additional forcing term is included in the model equations that controls the mean flow across the domain. The additional forcing term is a simple relaxation towards the vanishing mean flow, that is,


$$\boldsymbol{f} = (-\langle u_x \rangle/\tau, \ -\langle u_y \rangle/\tau, \ -\langle u_z \rangle/\tau \ ), \qquad (5)$$

where $\boldsymbol{u} = (u_x, \ u_y, \ u_z)$, $\langle . \rangle$ is the 3D average, and $\tau$ is the relaxation time scale taken as 10 model time steps. Note that (5) does not dump the flow perturbations, but only prevents the mean flow

development. In simulations presented here, the mean flow across the domain after applying (5) was limited to about $10^{-16}$ m s$^{-1}$.

Second, although (3) maintains the mean TKE, the TKE partitioning between the three velocity components is allowed to evolve. As a result, the magnitude of the root mean square (rms)

vertical velocity can vary in time and affect supersaturation fluctuations and droplet growth. Thus, (3) needs to be modified to maintain the uniform TKE partitioning between the three components. The idea is to apply the forcing for each component separately and assuming equipartition of the TKE between all three components. For instance, for the $x$ velocity component, $u_x$, the modified forcing should be $u_x^{(n+1)} = \alpha_x \, u_x^{(n)}$ with $\alpha_x = [2 \, E_t/3 \, \langle u_x^{(n) \, 2} \rangle]^{1/2}$,

where, as before, $E_t$ is the target TKE. The forcing term for $u_x$ is then given by:

$$f_x = u_x^{(n)}(\alpha_x - 1) \, / \, \Delta t \ , \qquad (6)$$

and similar for the two other velocity components.


To document the necessity of using the more elaborate approach (6), two simulations applying either (3) combined with (5) or (6) combined with (5) were run (details of the simulations are provided in the next section). Figure 1 compares TKE and rms vertical velocity in the two simulations. The figure shows that the modified forcing, that is, applying (6) in place of (3),

maintains not only the TKE, but also a uniform in time rms vertical velocity. The latter implies even partitioning of the TKE between the velocity components in agreement with the forcing formulation. The fluctuating in time rms vertical velocity in simulations applying forcing (3) leads to evolving supersaturation standard deviation in moist simulations and thus more complex mean droplet size evolution (not shown). In summary, the forcing term driving the isotropic

homogeneous turbulence applied in this study is the sum of (5) and (6).

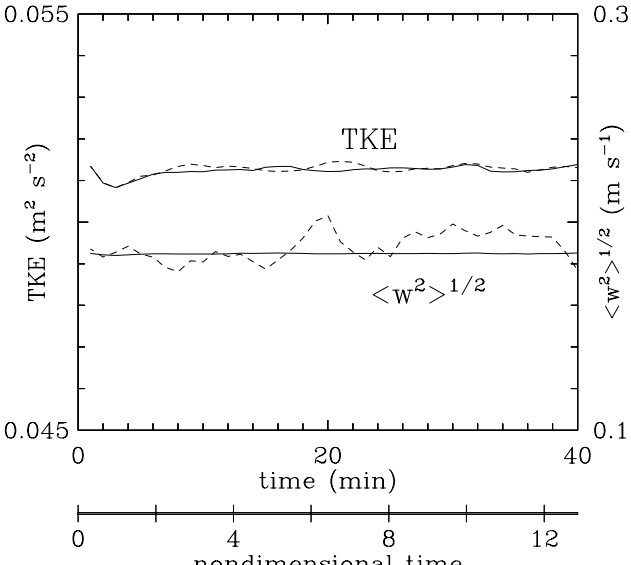


**Figure 1. Evolutions of the TKE and rms vertical velocity in low TKE simulations applying either (3) – dashed lines, or (6) – solid lines, as part of the forcing. The nondimensional time is the time divided by the turbulence integral time, 187 sec for the low TKE simulations.**


## 2.1 The setup of dynamic simulations

The triply-periodic computational domain is $64^3$ m$^3$ with the model grid length of 1 m. This is one of the domains considered in Thomas et al. (2020) and close to the turbulent rising parcel extent of 50 m considered in Grabowski and Abade (2017). Such a domain size is also similar to the grid volume of LES simulations of natural clouds (e.g., Siebesma et al., 2003, Stevens et al., 2005, VanZanten et al., 2011). For the fluid flow, we consider two turbulence intensities as expressed by the prescribed TKE. The "low TKE" simulations assume the TKE of 5.2 x 10$^{-2}$ m$^2$ s$^{-2}$. Such a TKE corresponds to the TKE dissipation rate of 10$^{-3}$ m$^2$ s$^{-3}$ in $64^3$ m$^3$ scaled-up DNS simulations in Thomas et al. (2020) that followed DNS simulations in Lanotte et al. (2009). The low TKE setup corresponds to the rms vertical velocity around 0.2 m s$^{-1}$ (see Fig. 1), evolving maximum vertical velocity between 0.5 and 0.8 m s$^{-1}$, and integral time scale (see Eq. 7 in Grabowski and Abade, 2017) of 187 sec, or about 3 minutes. The model time step for the low TKE simulations dictated by the CFL stability criterium is $\Delta t$ = 0.25 sec. The low TKE dry dynamics simulations (i.e., solving only Eqs. 1 and 2 only) and simulations without droplets (section 3) were run for 40 minutes (around 12 integral time scales as shown in Fig. 1). Simulations with droplets discussed in section 5 were run for 20 min or about 6 integral time scales starting from minute 28 of 40-min simulations presented in the next section. The "high TKE" simulations consider hundred times larger TKE, i.e., 5.2 m$^2$ s$^{-2}$. High TKE simulations feature ten times larger rms and maximum velocities (i.e., around 2 m s$^{-1}$ and 5 to 8 m s$^{-1}$, respectively) together with ten times smaller integral time scale of about 19 sec. Model time step in high TKE simulations was proportionally reduced to 0.025 sec. The high TKE simulations were run for the same number of time steps as the low TKE simulations, that is, for the total time of either 4 or 2 minutes, that is, either 12 or 6 integral time scales. Model data were saved every 15/1.5 sec for low/high TKE simulations and they are used in the analysis presented here.

Eq. 4 allows estimation of the TKE dissipation rate. The parameter $\alpha$ in the forcing term is monitored from time step to time step during the simulations. The typical value in both low and high TKE is $\alpha$-1 $\approx$ 2 x 10$^{-4}$. With the target TKE $E_t$ = 5.2 x 10$^{-2}$ m$^2$ s$^{-2}$ and $\Delta t$ = 0.25 sec in low TKE simulations, the TKE dissipation diagnosed from (4) is $\varepsilon \approx$ 4 x 10$^{-4}$ m$^2$ s$^{-3}$. This value approximately agrees with the assumed low TKE turbulence simulation setup. For the high TKE, the target TKE and the model time step imply $\varepsilon \approx$ 4 x 10$^{-1}$ m$^2$ s$^{-3}$. This is a rather extreme TKE

dissipation rate for small cumulus dynamics (e.g., Siebert et al., 2006), but perhaps not unusual for deep convection as simulated, for example, by Benmoshe et al. (2012).


Figure 1, already mentioned in section 2a, documents TKE evolution together with the rms vertical velocity in the low TKE simulation. The horizontal axis shows either the real time or the nondimensional time using the integral time scale. As the figure shows, the forcing maintains the TKE and rms vertical velocity as expected by the forcing design. Figure 2 shows the TKE

spectra for low and high TKE at the minute 28/2.8 that are used as initial conditions for moist simulations with droplets. The spectra have a classical shape characteristic of a relatively low-Reynolds-number homogeneous isotropic numerical turbulence (e.g., Fig. 2 in Rosales and Meneveau, 2005). Spectra at different times are similar to those in Fig. 2 (not shown).

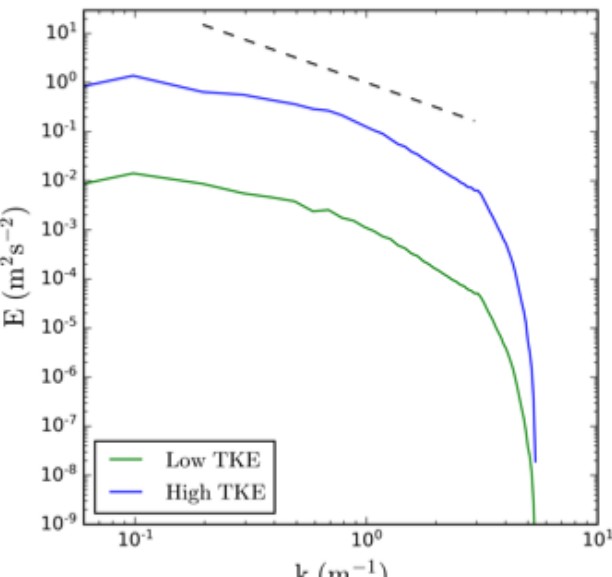


**Figure 2. Energy spectra for the fluid flow simulations without droplets at minute 28/2.8 for low/high TKE simulation without droplets. The dashed line represents the -5/3 Kolmogorov slope.**

The turbulence dynamics in moist simulations described in the next section is exactly as described above, that is, the impact of cloud droplets and of the latent heating on the flow is neglected. This is because the air density is assumed constant and the flow equations exclude the buoyancy term as typical in the homogeneous isotropic DNS simulations (see, for instance, eq. 1

in Lanotte et al., 2009). Because the flow is exactly the same in all simulations, one can compare model results grid volume by grid volume as in the piggybacking methodology (Grabowski, 2019 and references therein). This allows a comprehensive comparison of simulation results as illustrated in section 6.

**3 Thermodynamics in ILES moist simulations**

In addition to the momentum, the moist ILES of homogeneous isotropic turbulence solves the temperature $T$ and water vapor mixing ratio $q_v$ equations in the form:

$$\partial T/\partial t + \mathrm{div}\,(\boldsymbol{u}\,T) = L_v/c_p\,C_d - g/c_p\,u_z \quad, \quad (7)$$

$$\partial q_v/\partial t + \mathrm{div}\,(\boldsymbol{u}\,q_v) = -\,C_d \quad, \quad (8)$$

where $L_v = 2.5 \times 10^6$ J kg$^{-1}$ is the latent heat of condensation, $c_p = 1015$ J kg$^{-1}$ K$^{-1}$ is the specific heat of air at constant pressure, $g = 9.81$ m s$^{-2}$ is the gravitational acceleration, and $C_d$ is the condensation rate, the rate of change of the cloud water mixing ratio resulting from the diffusional growth of cloud droplets. Calculation of the condensation rate depends on the microphysics scheme as explained below and documented in the Appendix B.

In the spirit of DNS studies of homogeneous isotropic turbulence, the initial temperature and water vapor mixing ratio in moist simulations are assumed spatially uniform. The actual values are taken as in Thomas et al. (2020), that is, $T = 283$ K and $q_v$ at saturation assuming environmental pressure of 1000 hPa. The spatially-uniform initial conditions justify the triply-periodic computational domain. The last term in (6) represents the temperature change due to adiabatic air expansion resulting from the vertical motion in the stratified environment. This term drives small-scale supersaturation fluctuations in the otherwise uniform environment; see discussion in section 3 in Vaillancourt et al. (2001). Such a modeling framework is a simplification of a truly stratified environment where the environmental temperature and pressure are functions of height and typically the potential temperature (an invariant for dry adiabatic vertical displacements) rather than the temperature is applied as the model variable. In

some DNS studies, the temperature and moisture equations are combined into the supersaturation

equation that includes the source due to the vertical motion (as the last term in Eq. 6) and the

sink due to droplet growth, Eq. (2) in Lanotte et al. (2009) or Eq. (2) in Sardina et al. (2015); see

Eq. (10) below.

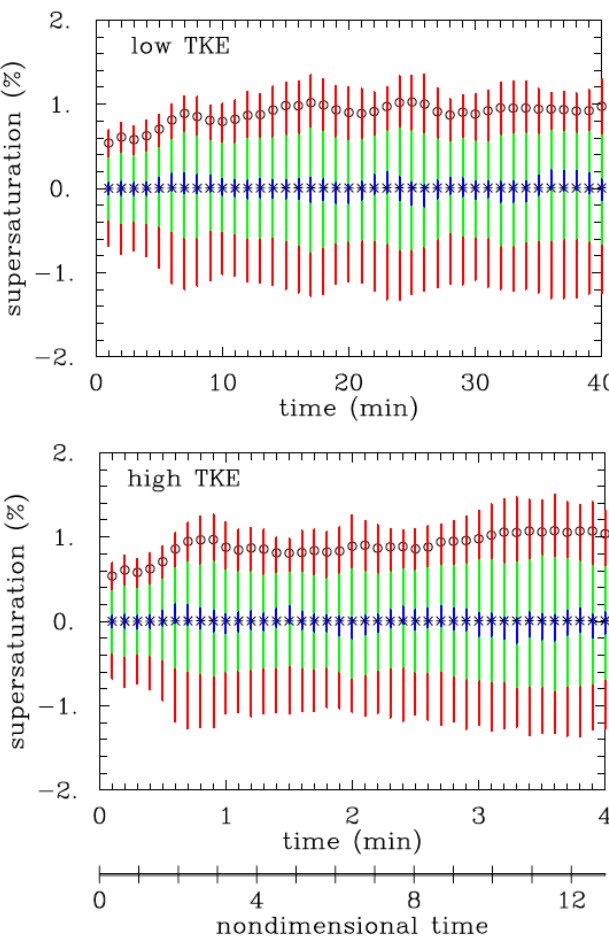


**Figure 3. Evolution of the supersaturation spatial distribution statistics for a 40-minute low TKE (upper panel) and high TKE (lower panel) moist simulations without droplets. Stars and circles show the mean and standard deviation of the spatial distribution, respectively. The extent of the color bars shows the percentiles of the distribution: red is for 10-90th percentile, green is for 25-75th percentile, and blue is for 45-55th percentile. The nondimensional time, the same for low**

**and high TKE, is shown below the lower panel.**

The initial test of the moist ILES framework considers a 40-min long low TKE and 4-min long

high TKE simulations without droplets (i.e., both up to about 12 integral time scales) and

initiated in the same way as the dry simulation illustrated in Figs. 1 and 2. The simulations apply

the fluid flow as described above and solving (7) and (8) without the condensation term $C_d$. In

such simulations, the largest temperature change is possible when the air parcel rises across the entire computational domain depth, that is, 64 m, with the corresponding temperature change of about 0.64 K as given by (7). Such a maximum temperature change leads to the supersaturation change from the initial zero to about 4%. In the numerical simulation, the maximum temperature deviations from the uniform initial 283 K are typically smaller than 0.5 K. The evolutions of supersaturation statistics are shown in Fig. 3. Despite the dramatic difference in the TKE levels, the statistics are similar regardless of the TKE level, in agreement with the parcel argument. Only after including the source due to condensation, the evolutions become different depending on the droplet characteristics and the TKE level. Small differences in the evolutions in Fig. 3 come from different flow realizations between low and high TKE cases.

**4 ILES moist simulations with droplets**

The general microphysical setup considers initially monodisperse population of cloud droplets with the radius of 13 μm present in three different concentrations: 26, 130, and 650 cm$^{-3}$. The concentration 130 cm$^{-3}$ was considered in Lanotte et al. (2009) and Thomas et al. (2020) and it corresponds to the mean cloud water content of about 1.2 g m$^{-3}$. In addition, the five-times smaller and five-times larger droplet concentrations are considered to document how the two microphysics schemes represent the expected scalings. The mean cloud water content in simulations with the increased (decreased) droplet concentration is about 6.0 (0.24) g m$^{-3}$. Moist simulations with droplets start at minute 28 of the moist simulation without droplets, that is, applying the flow field analyzed in Fig. 2, together with the temperature and moisture field at minute 28 in Fig. 3. The simulations are run for additional 20 minutes for the low TKE setup, saving data every 15 sec. The high TKE simulations are run for 2 minutes and the data are saved every 1.5 sec. The data are used in the analysis of both macrophysical (e.g., the supersaturation characteristics) and microphysical (e.g., droplet spectra) simulated by the two microphysics schemes. Sardina et al. (2015) derived scalings for the case when the droplet phase relaxation time is much shorter than the turbulence integral time scale. This is the case for the selected domain size and the low TKE for all droplet concentrations. For the high TKE and low droplet concentration (i.e., 26 cm$^{-3}$), the phase relaxation time (about 10 seconds) is the closest to the

turbulence integral time scale (about 19 seconds), so some deviations from the theoretical scaling should be expected as shown below.

## 4.1 Lagrangian and Eulerian microphysics schemes


For moist simulations with droplets, Eqs. (7) and (8) are supplemented by the appropriate equations describing droplet spectral evolution. The Lagrangian scheme follows evolution of so-called superdroplets, each representing an ensemble of real droplets, with the ensemble size referred to as the multiplicity. The bin scheme applies the Eulerian spectral density function

discretized into a finite number of radius bins. Details of both approaches are presented in the Appendix B.

The particle-based Lagrangian scheme considers on average 40 Lagrangian particles (superdroplets) per grid volume, each featuring the initial radius of 13 μm. The number of

superdroplets per grid volume together with the assumed droplet concentration dictates the multiplicity that is assumed the same for all superdroplets. Although the average number of superdroplets per grid volume is small when compared to millions real droplets within a 1 cubic meter grid volume, G20a and G20b document that the number as small as 10 per grid box provides physically-meaningful results; see also Li et al. (2017). At the simulation onset, each

superdroplet is placed at a random position within a grid volume. To be consistent with the bin microphysics, superdroplets grow in response to the mean supersaturation predicted inside a grid volume it occupies. Superdroplets are advected applying a model flow field interpolated to the droplet position as in Arabas et al. (2015). The interpolation scheme maintains the incompressibility of the flow at subgrid scales; see discussion of this aspect in section 2.4 in

Grabowski et al. (2018). Droplet inertia and droplet sedimentation are not considered. Condensation rate $C_d$ at each time step is calculated by summing up the mass change of all superdroplets present within a given grid volume.

The simulations with superdroplets are referred to as SDS.26, SDS.130, and SDS.650 for droplet

concentrations of 26, 130, and 650 cm$^{-3}$, respectively. The three simulations are completed for both low and high TKE. In addition, a single simulation with 150 superdroplets per grid volume

and droplet concentration of 130 cm$^{-3}$ was completed for the low TKE to test the impact of the superdroplet number fluctuations within a grid volume. This simulation is referred to as SDS.HR.130 (HR for high resolution in the radius space).


The Eulerian bin microphysics considers spectral density function represented by 40 equally-spaced bins with the bin size modified in different simulations as described below. The reason for modifications of the bin resolution is to match the results from the Lagrangian microphysics as shown in the results section. In each bin setup, there is a bin centered at 13 μm that is filled

with droplets at the simulation onset. The monodisperse initial droplet size distribution is impossible to be accurately represented using the spectral density function because the monodisperse distribution corresponds to the delta function. However, even with a finite width of the initial distribution, the broadening of the distribution as time progresses (Sardina et al., 2015; Li et al., 2019; Thomas et al., 2020) can be appropriately represented provided that the bin

width is appropriately small (see section 5.3). In the bin microphysics, each bin is independently advected in the physical space using the same advection scheme that is applied to the momentum, temperature, and water vapor mixing ratio. Neither droplet sedimentation nor droplet inertia are considered as in the Lagrangian scheme. All bins are combined at each grid volume to calculate evolution of the droplet spectrum due to the local sub- or supersaturation

applying a custom-designed 1D advection scheme. The scheme combines the analytic Lagrangian solution of the condensational growth with remapping of the spectral distribution onto the original radius grid using piecewise linear functions (see section 3.2 in Grabowski et al., 2011). As for the Lagrangian scheme, condensation rate $C_d$ at each time step is calculated from the change of the spectral density function due to the droplet growth in each grid volume, see

Appendix B.

For the low TKE, eight bin simulations with the spectral density function represented by 40 bins and different bin resolutions are considered. The selection of a specific bin resolution is motivated by the results discussed in the next section. The standard bin setup is similar to G20b

with a uniform 0.5 μm bin width and 0 to 20 μm bin range. These simulations are BIN.26, BIN.130, and BIN.650 for the three droplet concentrations. The high resolution (HR) simulations have 0.3 μm bin width and the bin layout centered at 13 μm (i.e., from 7 to 19 μm). These

simulations are run for 130 and 650 cm$^{-3}$ concentrations and are referred to as BIN.HR.130 and
BIN.HR.650, respectively. Very high resolution (VHR) simulations have 0.1 μm bin width,
again centered at 13 μm (bin range between 11 and 15 μm) and with 130 and 650 cm$^{-3}$
concentration, BIN.VHR.130 and BIN.VHR.650. Finally, an even higher bin resolution, 0.05 μm
bin width and grid centered at 13 μm (i.e., bin range between 12 and 14 μm), is added for the 650
cm$^{-3}$ concentration, BIN.SHR.650 (SHR for Super High Resolution). For the high TKE, only
three simulations were completed, BIN.26, BIN.VHR.130, and BIN.VHR.650. Table 1 provides
a list of all simulations in both Lagrangian and Eulerian simulations. Results of additional
simulations with a smooth initial droplet spectrum are presented in the Appendix A.

**Table 1. Details of Lagrangian and Eulerian simulations.**

Lagrangian (superdroplet) simulations:

| Low TKE: | SDS.26 | 26 cm$^{-3}$, 40 superdroplets per grid volume |
| | SDS.130 | 130 cm$^{-3}$, 40 superdroplets per grid volume |
| 445 | SDS.HR.130 | 130 cm$^{-3}$, 150 superdroplets per grid volume |
| | SDS.650 | 650 cm$^{-3}$, 40 superdroplets per grid volume |
| | | |
| High TKE: | SDS.26 | 26 cm$^{-3}$, 40 superdroplets per grid volume |
| | SDS.130 | 130 cm$^{-3}$, 40 superdroplets per grid volume |
| 450 | SDS.650 | 650 cm$^{-3}$, 40 superdroplets per grid volume |

Eulerian (bin) simulations:

| 455 | Low TKE: | BIN.26 | 26 cm$^{-3}$, 40 bins centered at 13 μm, 0.5 μm bin width |
| | | BIN.130 | 130 cm$^{-3}$, 40 bins centered at 13 μm, 0.5 μm bin width |
| | | BIN.HR.130 | 130 cm$^{-3}$, 40 bins centered at 13 μm, 0.3 μm bin width |
| | | BIN.VHR.130 | 130 cm$^{-3}$, 40 bins centered at 13 μm, 0.1 μm bin width |
| | | BIN.650 | 650 cm$^{-3}$, 40 bins centered at 13 μm, 0.5 μm bin width |
| 460 | | BIN.HR.650 | 650 cm$^{-3}$, 40 bins centered at 13 μm, 0.3 μm bin width |
| | | BIN.VHR.650 | 650 cm$^{-3}$, 40 bins centered at 13 μm, 0.1 μm bin width |
| | | BIN.SHR.650 | 650 cm$^{-3}$, 40 bins centered at 13 μm, 0.05 μm bin width |
| | | | |
| 465 | High TKE: | BIN.26 | 26 cm$^{-3}$, 40 bins centered at 13 μm, 0.5 μm bin width |
| | | BIN.VHR.130 | 130 cm$^{-3}$, 40 bins centered at 13 μm, 0.1 μm bin width |
| | | BIN.VHR.650 | 650 cm$^{-3}$, 40 bins centered at 13 μm, 0.1 μm bin width |

470    Droplet growth in both schemes is calculated applying a simplified growth formula as in G20a
and G20b:

$dr/dt = A \, S / (r + r_0),$  (9)

with $A = 0.9152 \times 10^{-10}$ m$^2$ s$^{-1}$ and $r_0 = 1.86$ μm. The latter is applied to mimic the impact of kinetic effects (Mordy 1959, see Eq. 11 in Clark 1971 or Eq. 2.22 in Kogan 1991). Because of a large mean droplet size, 13 μm, the solution and curvature effects are neglected. The two schemes apply the droplet growth equation (9) in different ways: as a transport (advection) velocity in the Eulerian bin scheme and to calculate individual superdroplet growth in the Lagrangian scheme, see Appendix B.

**5. Results**

**5.1 Lagrangian microphysics**

Figure 4 shows evolutions of the supersaturation standard deviation in low and high TKE Lagrangian simulations. The supersaturation standard deviation is approximately constant except for the adjustment from initial values in simulations without droplets (cf. Fig. 3). Standard deviations for the SD.130 and SD.HR.130 simulations are practically the same. Table 2 shows the standard deviation averaged over the second half of the simulations. When the phase relaxation time is much smaller than the turbulence integral time scale, the supersaturation standard deviation is proportional to the product of the rms vertical velocity and the phase relaxation time; $\sigma_S \sim \langle w^2 \rangle^{1/2} \, \tau_{relax}$, see Sardina et al. (2015). The phase relaxation time is inversely proportional to the product of the mean droplet radius and droplet concentration. With the same mean droplet radius, changes in the concentration explain the factor of about five shifts between SDS.650, SDS.130, and SDS.26 for the low TKE case as shown in the left panel and in Table 2. However, for the high TKE, the scaling breaks down because the phase relaxation time is no longer much smaller than the turbulence integral time scale. For a given droplet concentration, shift from low to high TKE should result to ten-fold increase of $\sigma_S$ because of the $\langle w^2 \rangle^{1/2}$ increase. This is approximately valid for 650 cm$^{-3}$ droplet concentration, but reduces to a factor of only about 5 for 26 cm$^{-3}$ concentration.

The supersaturation standard deviation shown in Fig. 4 can be compared to the standard deviation resulting from the quasi-equilibrium supersaturation fluctuations. Evolution of the

supersaturation $S = q_v/q_{vs} - 1$ (where $q_{vs}$ is the saturated water vapor mixing ratio) can be derived by combining Eqs. 7, 8 and 9 as

505    $dS/dt = a_1 w - S/\tau_{relax}$   ,       (10)

where $\tau_{relax}$ is the phase relaxation time that depends on the mean droplet radius and concentration:

510    $1/\tau_{relax} = 4 \pi \rho_w A [1/q_{vs} + q_v L_v^2/(q_{vs} R_v T^2 c_p)] <N r^2/(r+r_0)>$ ,   (11)

where $\rho_w = 10^3$ kg m$^{-3}$ is the water density, $R_v = 461$ J kg$^{-1}$ K$^{-1}$ is the water vapor gas constant, and $<.>$ in (11) depicts averaging over all droplets within a given grid volume. The quasi-equilibrium supersaturation is obtained by setting the left-hand-side of (10) to zero that leads to

515    $S_{eq} = a_1 w \tau_{relax}$ . For the mean temperature and humidity of the simulations and specific numerical values of the relevant constants, $a_1 = 6.54$ x $10^{-4}$ m$^{-4}$ and the phase relaxation time for the 13 μm droplets and their concentration of 130 cm$^{-3}$ is $\tau_{relax} = 1.98$ sec. The phase relaxation time is five times smaller/larger for the droplet concentration of 650/26 cm$^{-3}$. Since the quasi-equilibrium supersaturation is proportional to the vertical velocity, its standard deviation is

520    proportional to the rms vertical velocity. For the low TKE, the quasi-equilibrium supersaturation standard deviation is 0.120, 2.39e-2, 4.79e-3% for 26, 130 and 650 cm$^{-3}$ droplet concentrations. These are in a relatively good agreement with simulated standard deviation shown in Table 2. The values for the high TKE should be 10 times smaller and they are approximately equal to the simulated values for 130 and 650 cm$^{-3}$. The agreement between the simulated supersaturation

525    fluctuations and the fluctuations predicted by the quasi-equilibrium supersaturation for the 64$^3$ m$^3$ domain and low TKE agrees with results presented in Thomas et al. (2020; see Fig. 10 therein and its discussion).

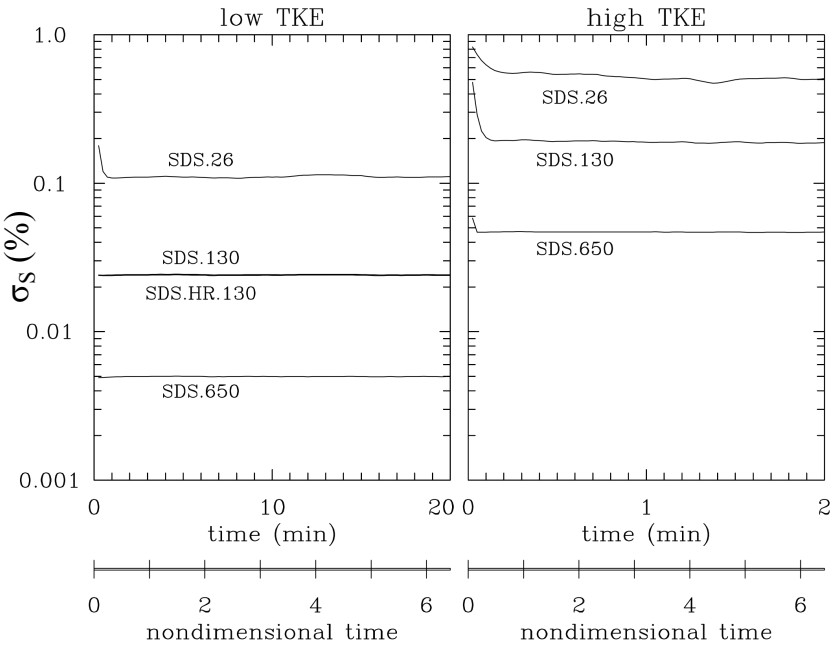

**Figure 4. Evolution of the supersaturation standard deviation in (left panel) 4 low TKE simulations and (right panel) 3 high TKE simulations with superdroplets. The two simulations with droplet concentration of 130 per cc in the left panel, SDS.130 and SDS.HR.130, show differences within the thickness of the line.**

Figure 5 shows the comparison between the local supersaturation simulated by the model and the quasi-equilibrium supersaturation calculated applying the local vertical velocity and the mean phase relaxation time for the low and high TKE simulations and 26 versus 650 cm$^{-3}$ droplet concentrations. The mean phase relaxation time is about 10 sec for 26 cm$^{-3}$ and about 0.4 sec for 650 cm$^{-3}$ droplet concentrations. Whether the quasi-equilibrium supersaturation is a good approximation of the local supersaturation depends on the relative magnitude of the phase relaxation time scale and the eddy turnover time associated with the largest eddies. This is because the largest eddies feature the largest vertical velocities and provide the strongest forcing to drive the supersaturation away from its quasi-equilibrium value. For the low TKE, the eddy turnover time for the largest eddies is about 1 minute (i.e., velocities up to 0.8 m s$^{-1}$ and the domain size of 64 m), much larger than the phase relaxation time for both concentrations shown in Fig. 5. This is why all points scatter around 1:1 line in the left panels. However, the eddy turnover time is only around 8 sec for the high TKE case. This is still much larger than the phase relaxation time for the 650 cm$^{-3}$ droplet concentration (lower right panel), but close to the phase relaxation time for the 26 cm$^{-3}$ concentration. This is why data points are scattered away from the

1:1 line in the upper right panel, with the quasi-equilibrium values typically larger (in the absolute sense) than the model-predicted supersaturation.

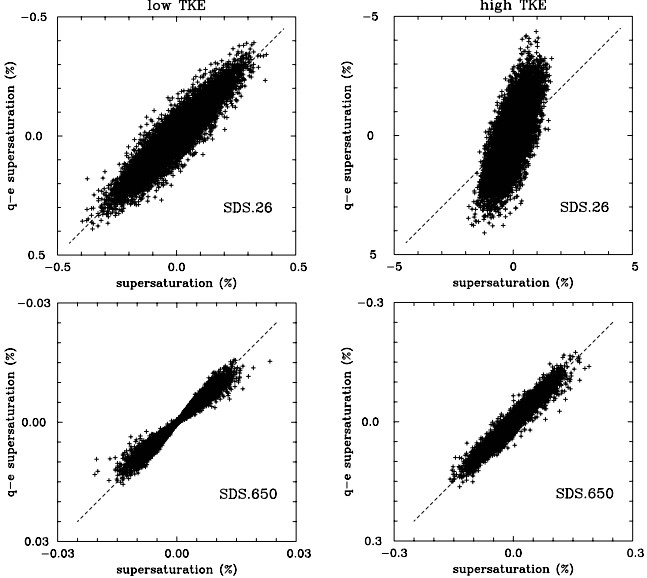

**Figure 5. Comparison between supersaturation simulated by the model (horizontal axes) and the quasi-equilibrium supersaturation calculated with the local vertical velocity and the mean phase relaxation time (vertical axes) for (left panels) low TKE and (right panels) high TKE simulations. Simulations SDS.650/SDS.26 are in the lower/upper panels. Data from the last time level of all simulations with only 5% of data points shown. Note different supersaturation ranges in all panels.**

Figure 6 shows evolutions of the radius squared standard deviation. Sardina et al. (2015) show that the standard deviation of the radius squared distribution should increase in time as square root of time as long as the phase relaxation time is much smaller than the turbulence integral time. The rate of increase is proportional to the supersaturation standard deviation (see Eq. 13 therein). The former has been shown in other numerical simulations, such as in Li et al. (2019) and Thomas et al. (2020). Fig. 6 shows that the Lagrangian microphysics reproduces the $t^{1/2}$ scaling and that the differences between various simulations for the low TKE can be explained by the differences in the supersaturation standard deviation shown in Fig. 4 (note that on the log-log plot the rate of increase change corresponds to a vertical shift as shown in Fig. 6). For the high TKE, small deviations for the expected scaling can be explained by the phase relaxation time being no longer much smaller than the turbulence integral time. This is especially evident for the high-TKE SDS.26 case in the right panel of Fig. 6. Left panels of Figs. 4 and 6 show that

increasing the number of superdroplets from 40 to 150 per grid volume has virtually no impact
on the results. Overall, the Lagrangian microphysics seems to represent expected scalings (or
departures from them) without much difficulty.

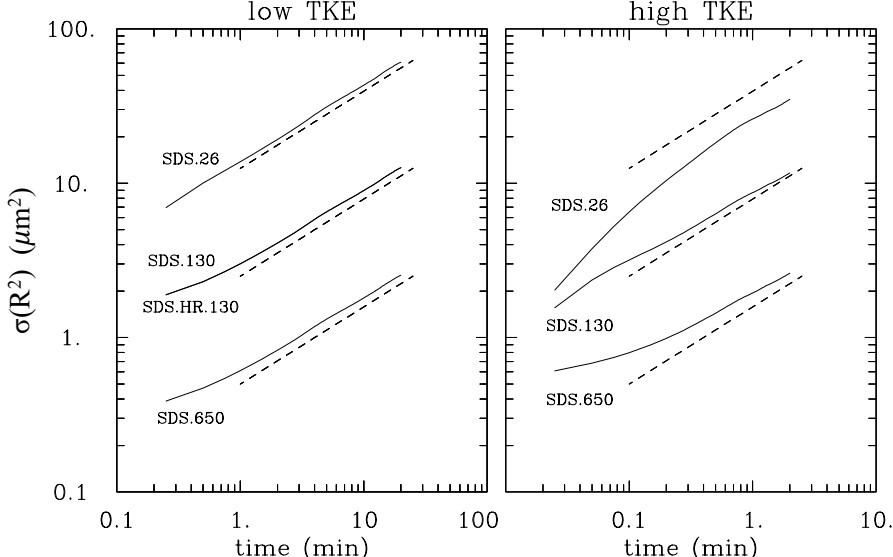


**Figure 6. Evolution of the radius squared standard deviation for superdroplet simulations (left panel) 4 low TKE simulations and (right panel) 3 high TKE simulations with superdroplets. SDS.130 and SDS.HR.130 in the left panel differ by the thickness of the line. Dashed lines show expected $t^{1/2}$ scaling and are spaced by the expected factor of 5. Their position is the same in left and right panels.**


**Table 2. Supersaturation standard deviation (%) averaged over the last 3 integral time scales for Lagrangian microphysics simulations.**

|  | N=26 | N=130 | N=650 |
|---|---|---|---|
| Low TKE | 0.111 | $2.42 \times 10^{-2}$ | $4.98 \times 10^{-3}$ |
| High TKE | 0.500 | 0.188 | $4.68 \times 10^{-2}$ |

## 5.2 Eulerian bin microphysics


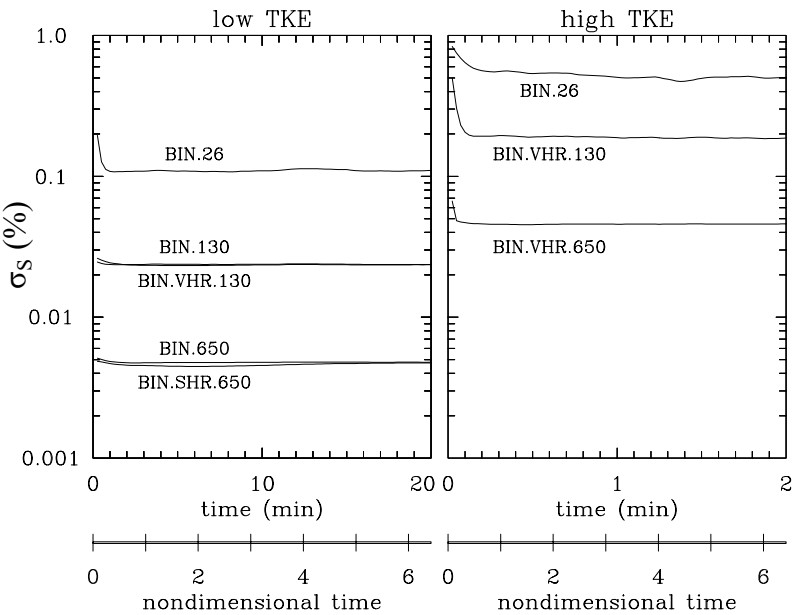

**Figure 7. As Fig. 4 but for the supersaturation standard deviation in 8 simulations with bin microphysics.**

Figures 7 and 8 show the same results as Figs. 4 and 6 for the bin microphysics. For the supersaturation fluctuations (Fig. 7), bin simulations match Lagrangian microphysics results, and the impact of bin resolution is small, at least for the low TKE simulations which feature various bin resolutions. However, as shown in Fig. 8, the expected $t^{1/2}$ scaling requires appropriately high bin resolution, and the resolution requirement changes depending of the droplet concentration.

The standard bin resolution is sufficient for the BIN.26 simulation. However, 130 cm$^{-3}$ simulations require VHR setup (bin width of 0.1 μm) to match the expected scaling. Even the SHR setup (bin width of 0.05 μm) is insufficient for the 650 cm$^{-3}$ droplet concentration. There are some similarities between Lagrangian and Eulerian results for the high-TKE simulations, that is, when the scalings derived by Sardina et al. (2015) may not apply. The left panel also shows

the decrease of the initial radius squared standard deviation with the increase of the bin resolution. This comes from the ill-posedness of the initially monodisperse droplet size distribution for the bin microphysics. The comparison between the local supersaturation predicted by the model and the quasi-equilibrium supersaturation calculated using the local vertical velocity (i.e., Fig. 5) for the bin microphysics is similar to that shown for the Lagrangian

scheme and is not shown.

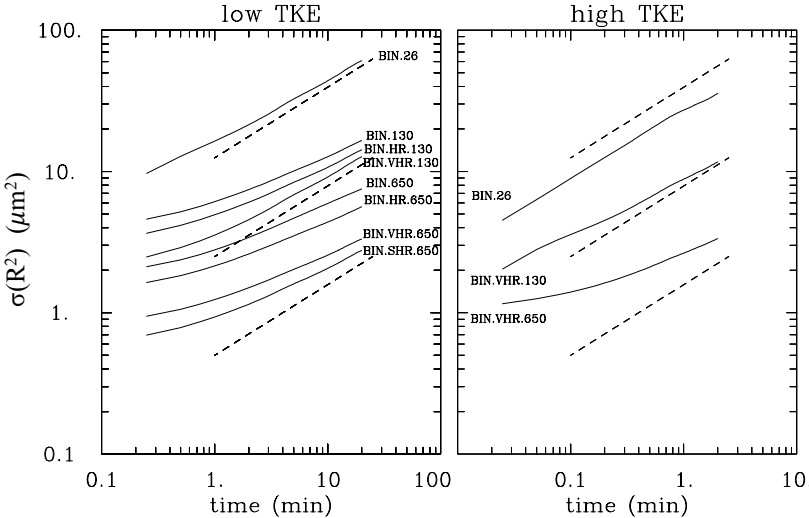

**Figure 8. Evolution of the radius squared standard deviation for bin simulations. Dashed lines show the expected $t^{1/2}$ scaling. Their positions are exactly as in Fig. 6 for the Lagrangian microphysics.**

In summary, Eulerian bin microphysics is capable in appropriately representing turbulent temperature and moisture fluctuations, but fails to simulate their impact on droplet spectra unless appropriately high bin resolution is used. This is further supported by the comparison of droplet spectra discussed in the next section.

## 5.3 Comparison of radius squared distributions between Eulerian and Lagrangian simulations.

This section compares radius squared ($R^2$) distributions at the end of the simulations, that is, after 6 turnover times, for both the low and high TKE simulations. As shown in Lanotte et al. (2009) and Sardina et al. (2015), an initial monodisperse distribution should evolve into a Gaussian $R^2$ spectrum because of the parabolic cloud droplet growth equation. Although the parabolic growth is only approximately valid because of the specific droplet growth equation (see Eq. 9), the Gaussian distribution is a good fit for simulation results discussed here as shown below.

 Figure 9 shows the spectra for selected superdroplet simulations. The radius squared spectra are created by selecting $R^2$ bin size and binning superdroplet radii for a given simulation into the assumed bin grid. The bin size for the SDS.650/SDS.26 simulations (lower/upper panels in Fig.9) is 1/10 $\mu m^2$. There are two panels for each simulation, one with the linear vertical scale

and the spectrum shown as a histogram, and the second one with the logarithmic vertical scale and using star symbols to show the spectrum. In addition, the logarithmic plots show the Gaussian distributions obtained with the mean and standard deviation calculated from the spectra.


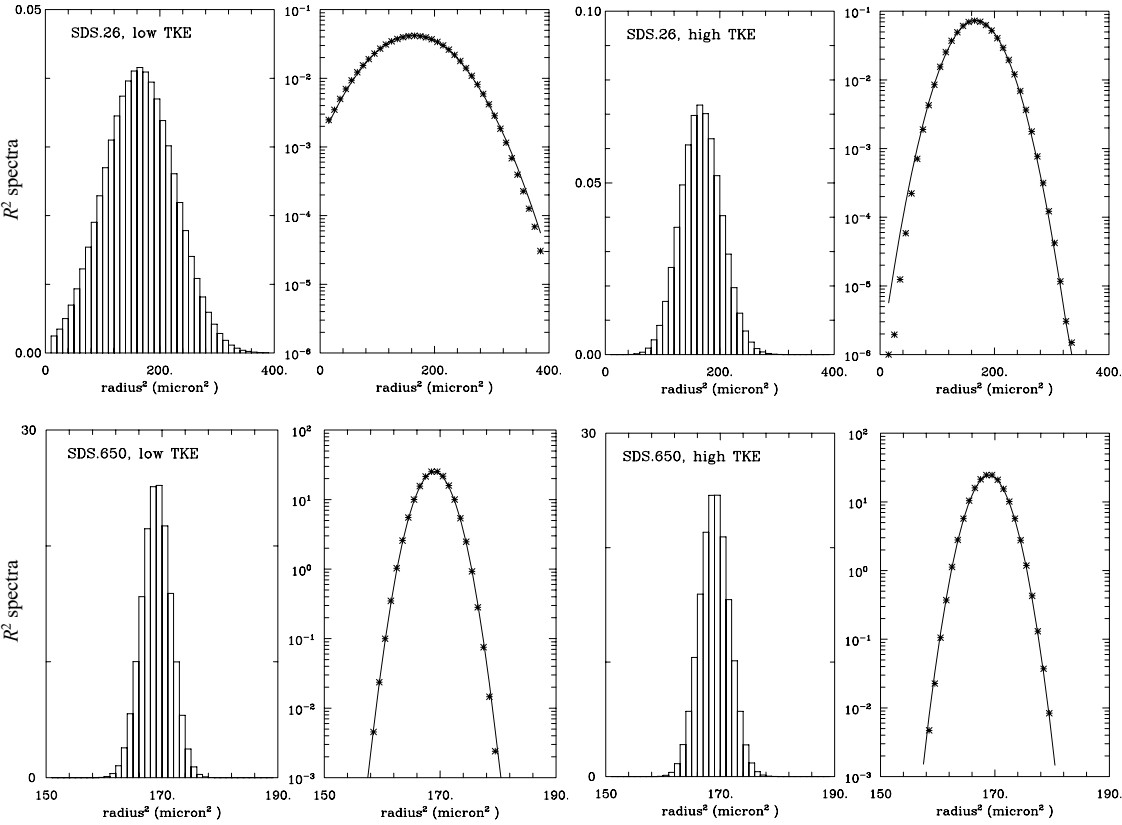

**Figure 9. Results from simulations (upper panels) SDS.26 and (lower panels) SDS.650 superdroplet simulations. There**
**are two panels for each simulation, the left one applying the linear vertical scale and the right one applying the logarithmic scale. The line in the logarithmic scale panels shows the Gaussian distribution with the mean and standard deviation calculated from the spectrum. Left/right pair in each row is for low/high TKE simulation.**

For the SDS.650 simulations (lower panels in Fig. 9), the spectra at the end of low and high TKE
simulations are practically the same. This agrees with the theoretical scaling and simulation results shown in Fig. 4 and 6. In contrast, results for SDS.26 differ drastically between the low and high TKE. The spectrum for the low TKE is wide, with some small droplets already evaporated because the spectrum is truncated at the low-radius end. Nevertheless, the Gaussian shape is still a good fit for the simulated spectrum. The high TKE SDS.26 spectrum is
significantly narrower with small deviations from the Gaussian fit.

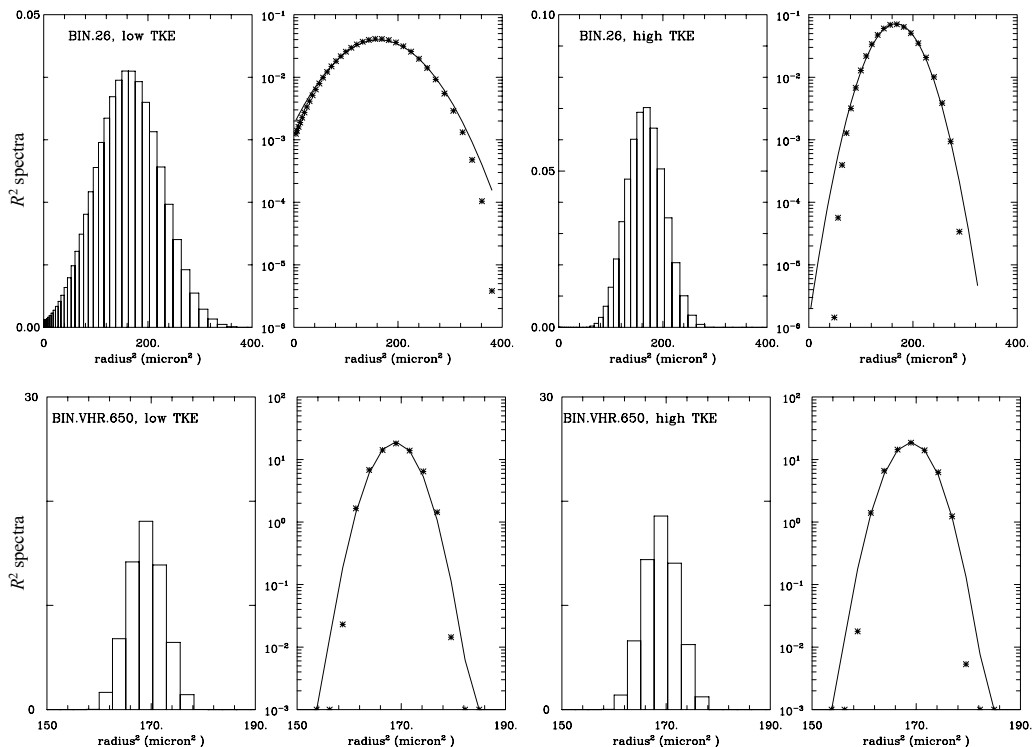

**Figure 10. As Fig. 9, but for the bin (upper panels) BIN.26 and (lower panels) BIN.VHR.650 simulations.**

Figure 10 shows the spectra for bin simulations similar to those in Fig. 9. Since bin simulations predict the spectra directly, the radius spectra are converted to $R^2$ spectra and then plotted at their native resolution in the $R^2$ space. This explains the change in the resolution along the horizontal axes evident in the upper panels. Overall, there are some similarities between Figs. 9 and 10. For instance, upper panels show spectra for the 26 cm$^{-3}$ simulations with 0.5-μm bin width that are similar to those in superdroplet simulations. Spectra for 650 cm$^{-3}$ simulations with 0.1-μm bin width (i.e., from the VHR set) are also similar between low and high TKE simulations, but their spectral widths are larger than in corresponding panels of Fig. 9. The impact of the bin resolution is further documented in Fig. 11 that shows results from the 650 cm$^{-3}$ low TKE HR and SHR simulations, that is, with the bin width of 0.3 and 0.05 μm, respectively. Only the SHR simulation (i.e., the right panel in Fig. 11) resembles the spectra from the Lagrangian simulations shown in the lower panels of Fig. 9.

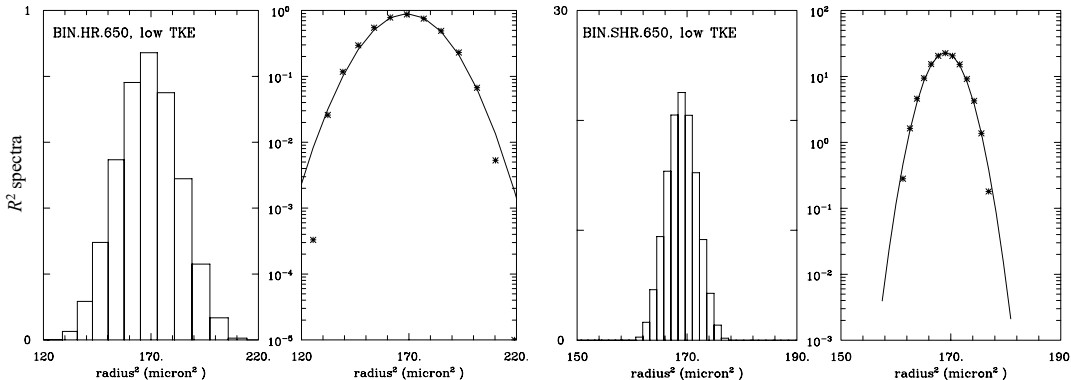

**Fig. 11. As Figs. 9 and 10, but for the bin BIN.HR.650 and BIN.SHR.650 simulations. Note different horizontal range at the left and right pair of panels.**


In summary, only extreme resolutions of the bin scheme (e.g., as in SHR, 0.05 μm bin width) allow good agreements between Lagrangian and Eulerian results for the concentration range considered here. Moreover, the ill-posed initial condition for the Eulerian scheme (i.e., the monodisperse initial droplet size distribution) seems irrelevant because the spectrum becomes

well-resolved after some time during the simulation. With sufficiently high bin resolution, (e.g., 0.5 μm in the 26 cm$^{-3}$ simulations or 0.05 μm for the 650 cm$^{-3}$ simulations), the Eulerian and Lagrangian spectra compare well at the end of the simulations. This shows the benefit of the Lagrangian scheme as one does not have to worry about the bin size to obtain numerically converged solutions.


**6 Grid volume by grid volume analysis of macro- and microphysical properties**

The analysis presented in previous sections concerns domain-averaged characteristics. Because all simulations with either low or high TKE feature exactly the same evolving flow field,

simulated thermodynamic variables (i.e., the temperature, water vapor, and cloud droplet characteristics) can be compared grid volume by grid volume and thus provide a comprehensive comparison of simulated local conditions. Such a comparison is in the spirit of the piggybacking methodology applied in G20b.

Figure 12 compares the temperature, water vapor, and cloud water mixing ratios (the latter derived from the predicted droplet spectra within each grid volume) between SDS.130 and BIN.130 low TKE simulations at time of 20 minutes. For the temperature and moisture, plots at

earlier times are similar to upper panels in Fig. 12 except for smaller ranges between minima and

maxima. Cloud water plots at earlier times are also similar to those shown in Fig. 12 except for

the initial couple minutes. The temperature and water vapor values are extremely close between

the two simulations: the root mean square difference between temperatures are $2.6 \times 10^{-4}$ K. For

the water vapor mixing ratios, the root mean square difference is $1.1 \times 10^{-4}$ g kg$^{-1}$. However, the

cloud water mixing ratio can differ significantly between the bin and the superdroplet

simulations. This is because of statistical fluctuations in the number of superdroplets per grid

volume. With on average 40 superdroplets per grid volume, the standard deviation of the droplet

number is around 6, or about 15% of the mean. Assuming that one can find grid volumes with

three times the standard deviation, the range of the cloud water mixing ratio can be as high as

close to 50%. This can explain the spread seen in the lower left panel of Fig. 12. With 150

superdroplets per grid volume in SDS.HR.130, there is some improvement as the standard

deviation is reduced to about 8% of the mean, but the statistical fluctuations remain significant.

In fact, the lower left panel does not change significantly if SDS.130 is replaced by SDS.HR.130

(not shown). The lower right panel shows the outcome of a simple rescaling of the cloud water

mixing ratio $q_c$ predicted by the Lagrangian scheme based on the number of superdroplets $N$

being present in a given grid volume compared to the expected mean value of 40, with the

rescaled cloud water mixing ratio given by $q_c$ $40/N$ (i.e., increased when $N < 40$ and reduced

when $N > 40$). We stress that the rescaling is done on the analyzed cloud water, and not during

the model run. (That said, application of such a rescaling might be a valuable approach to reduce

the spread during model run as well; this aspect is left for a future investigation). Apparently, the

rescaling improves the comparison significantly and documents that the scatter present in the

lower left panel comes predominantly from the statistical fluctuations in the Lagrangian scheme.

An important point is that the cloud water fluctuations are short-lived. This is because the

temperature and water vapor would feature larger scatter if the fluctuations in the grid volume

superdroplet number were long-lived. These statistical fluctuations come only from superdroplet

advection by the resolved flow as the inertial effects and droplet sedimentation are not

considered.

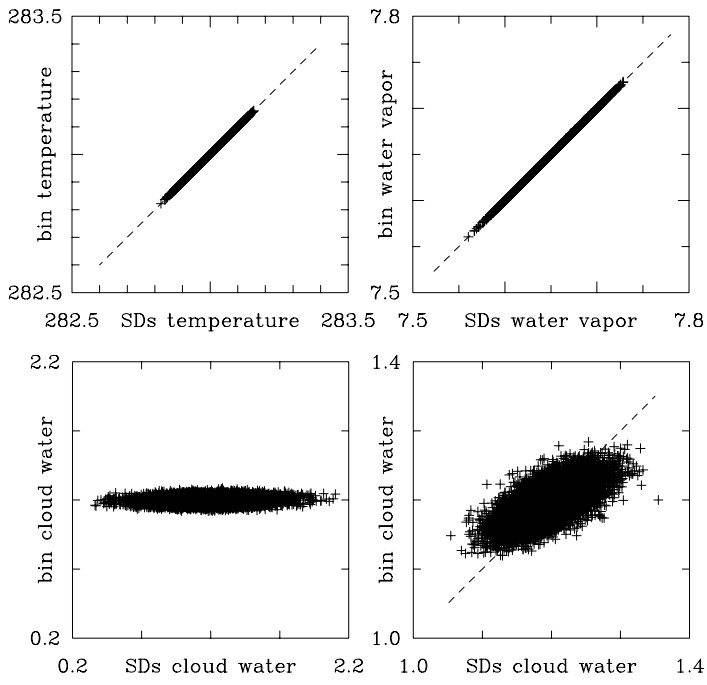


**Figure 12. Grid volume by grid volume comparison for the low TKE SDS.130 and BIN.130 simulations between (upper left) temperature (in K), (upper right) water vapor mixing ratio (in g kg⁻¹), and (lower left) cloud water mixing ratio (in g kg⁻¹) at minute 20. The lower right panel shows cloud water mixing ratio comparison after the SDS.130 results are adjusted as explained in the text. Note the change of horizontal and vertical scales between lower left and lower right**

**panels. Only 5% of data points are used.**

Figure 13, in the format of Fig. 12, compares the grid volume mean radii and spectral width in two sets of low-TKE simulations, the standard resolution (SDS.130 and BIN.130) and the increased resolution (SDS.HR.130 and BIN.VHR.130). For the increased bin resolution, the

selected bin microphysics is the one that shows the correct scaling in Fig. 8. The mean radius comparison features some scatter that is reduced when increased resolutions simulations are compared. However, the scatter is asymmetric with respect to 1:1 line and similar to the cloud water scatter in the lower right panel of Fig. 12. The asymmetry shows that the bin microphysics tends to simulate larger droplets than the Lagrangian microphysics for droplets smaller than the

mean, and the reverse is true for droplets larger than the mean. The spectral width panels show that the bin simulations feature smaller spread of the spectral width across the computational domain than the Lagrangian scheme. This seems independent of the bin resolution. In other words, spectral width simulated by the bin scheme varies less across the computational domain. In contrast, superdroplet simulations feature larger spread of the spectral width across the

domain, and the spread decreases with the increase of the mean superdroplet number per grid

volume. For the SDS.130 versus BIN.130 cloud of spectral width points (i.e., the upper right

panel), the center of mass is above the 1:1 line. This implies that the mean spectral width for

BIN.130 simulation is overpredicted, in agreement with the results shown in Fig. 8. Results for

similar comparisons of other simulations (for instance, SDS.26 versus BIN.26 or SDS.650 and

BIN.SHR.650) are similar except for different ranges of the mean radius and spectral width (not

shown). As shown in Fig. 14, the high TKE simulations also show similar patterns, with changes

consistent with the differences in right panels in Figs. 6 and 8.

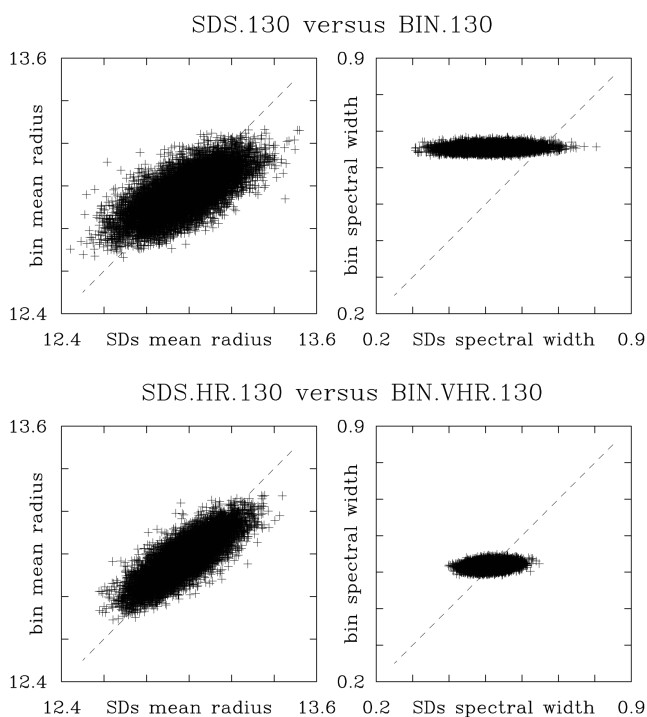


**Figure 13. Grid volume by grid volume comparison at the end of low TKE simulations (20 min) between (left panels) the grid-volume mean radius (in microns) and (right panels) spectral width (in microns). Lower (upper) panels are for comparison between SDS.HR.130 and BIN.VHR.130 (SDS.130 and BIN.130). Only 5% of data points are used.**


In summary, the grid volume by grid volume comparison between Eulerian and Lagrangian

results shows that the simulated spatial variability is smaller in the bin microphysics when

compared to the superdroplets. Arguably, this comes from a combination of the numerical

diffusion in the bin microphysics (i.e., smoothing bin results similar to other Eulerian fields) and

small-scale fluctuations of the Lagrangian microphysics due to a relatively small mean number

of superdroplets per grid volume.

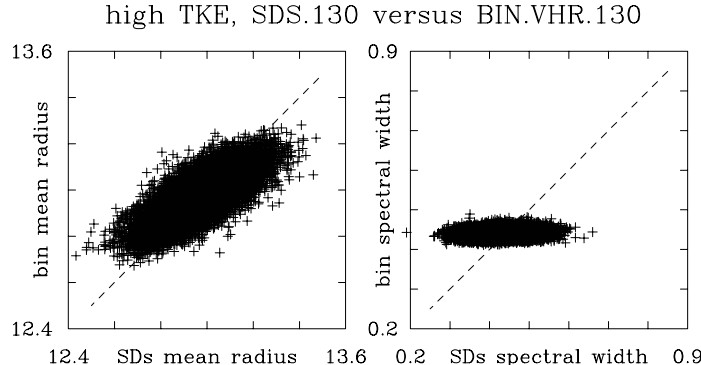

**Figure 14. As Fig. 10, but for SDS.130 and BIN.VHR.130 simulations with high TKE.**

## 7 Summary and conclusions

This paper presents a modeling study addressing the impact of homogeneous isotropic turbulence

on the broadening of initially monodisperse distribution of cloud droplets in response to local

fluctuations of the supersaturation field. This problem has been considered previously in

modeling studies of Lanotte et al. (2009) applying DNS, Thomas et al. (2020) using the scaled-

up DNS, and Sardina et al. (2015) employing theoretical analysis combined with DNS and

stochastic model simulations. Sardina et al. (2015) derived scaling relationships that we use in

validating model results and comparing results for different droplet concentrations and

contrasting turbulence intensities.

Because we apply a finite-difference fluid flow model, we had to develop a turbulence forcing

scheme that led to the quasi-steady homogeneous isotropic turbulence similar to that simulated

by a spectral model. The forcing scheme applied here is in the spirit of the linear forcing of

Rosales and Meneveau (2005) and Onishi et al. (2011). The idea is to ensure that the mean

turbulence kinetic energy (TKE) remains constant in time and that the partitioning between the

three TKE components does not change. The latter is important for the simulations of the

turbulence impact on the droplet spectra because the driving mechanism in the homogeneous

environment comes from the vertical velocity fluctuations affecting the local supersaturation. The simulations are in the spirit of the implicit large eddy simulation (ILES), that is, without modeling the small-scale TKE and scalar variance dissipation (Margolin and Rider, 2002; Andrejczuk et al., 2004; Margolin et al., 2006; Grinstein et al., 2007). ILES applying a finite difference model relies on the model numerics to provide the required dissipation, in contrast to

the scaled-up spectral model DNS in Thomas et al. (2020) where appropriately scaled-up molecular dissipation coefficients have to be used to ensure stable simulations. The TKE dissipation rate diagnosed from the forcing algorithm (see Eq. 6) is approximately correct for the considered turbulence intensities.

There are two drastically different simulation techniques that can be applied to investigate the impact of cloud turbulence on the droplet spectra: the Eulerian bin microphysics and the Lagrangian particle-based approach, the latter often referred to as the superdroplet method (see review in the introduction). We apply the ILES homogeneous isotropic turbulence setup to compare the two techniques following Li et al. (2017), Grabowski (2020a), and Grabowski

(2020b) for the diffusional growth of cloud droplets only. The computational domain is $64^3$ m$^3$, one of the domain sizes considered in Thomas et al. (2020) and similar to grid volumes of a typical LES simulation of natural clouds. Two TKE intensities are considered, low and high, different by a factor of one hundred. The latter implies that the velocity fluctuation differ by a factor of ten and the TKE dissipation rates differ by a factor of one thousand. The TKE

dissipation spans the range observed in natural clouds.

The Lagrangian approach reproduces the expected scalings derived in Sardina et al. (2015) for the case when the turbulence integral time scale is much longer that the phase relaxation time of cloud droplets. Representing the scalings is more challenging for the bin microphysics because

appropriately high resolution in the bin space is needed. In fact, the standard bin resolution, with the bin width of 0.5 μm and covering the range up to 20 μm, similar to Grabowski (2020a, 2020b), is only sufficient for the lowest droplet concentration (26 cm$^{-3}$). For the highest droplet concentration, 650 cm$^{-3}$, even an order of magnitude smaller bin size is not sufficient to reproduce well the expected scaling. Such a bin resolution is impossible to use when collisional

growth is also considered as in Li et al. (2017). For the lowest droplet concentration (26 cm$^{-3}$)

and the high TKE case, the phase relaxation time is about 10 sec and the turbulence integral time is around 19 sec, so some departures from the expected scaling are expected. This is indeed the case, and the two simulation methodologies represent similar supersaturation and spectral width departures.


Because the fluid flow is the same for all simulations featuring either low or high TKE, one can compare model results point-by-point as in the piggybacking technique of Grabowski (2019). Such a comparison shows miniscule differences between temperature and water vapor fields across the computational domain, and larger differences between simulated mean droplet radii and spectral width. These are consistent with fundamental differences in the two simulation methodologies, numerical diffusion in the Eulerian approach and relatively small number of Lagrangian particles (superdroplets) that can be afforded in the particle-based microphysics. Either one can be limited by either increasing the model resolution or increasing the number of Lagrangian particles, both significantly increasing computational cost. But there are additional options for the particle-based microphysics, for instance, assuming that a particle within a given grid volume represents a cloud of particles spread over a prescribed halo. We are pursuing those ideas in ongoing research.

**Appendix A. The impact of initial conditions.**


This appendix describes results from additional Eulerian bin and Lagrangian superdroplet low TKE simulations for the case of the total droplet concentration of 130 cm$^{-3}$ with the initial either monodisperse droplet spectrum (i.e., as in the main text) or a wide spectrum that can be well represented by the bin microphysics. For the latter, a truncated Gaussian spectrum is used with the width of 1 μm and truncated to zero for droplet radius outside the 10 to 16 μm range (i.e., three standard deviations). For the bin microphysics, the bin setup is as in the BIN.HR.130 (i.e., high-resolution) simulation, that is, with the bin width of 0.3 μm. With a single bin centered at 13 μm, there are 21 nonzero bins for the truncated Gaussian distribution in the 40-bin Eulerian setup. To simplify the super-droplet setup, we use 42 superdroplets per grid volume (rather than 40 as in main-text simulations), repeating twice the 21 superdroplet radii and multiplicities to exactly match the non-zero 21 bins in the initial bin setup. Motivated by the initial results, we

extend ten times the simulation length, that is, up to 200 min, over 60 eddy turnover times. The results are documented in Figs. A1 and A2.

Figure A1 shows evolution of the radius squared standard deviation in the format of Figs. 6 and 8 of the main text for simulations with monodisperse and with truncated Gaussian initial conditions. Extending ten times the length of simulations from the main text shows that the Lagrangian simulation continues on the expected $t^{1/2}$ scaling, and the bin simulation continues to approach that scaling. The truncated Gaussian initial droplet distribution simulations are close to

each other, and they start to approach the expected scaling only after the initial 10 min.

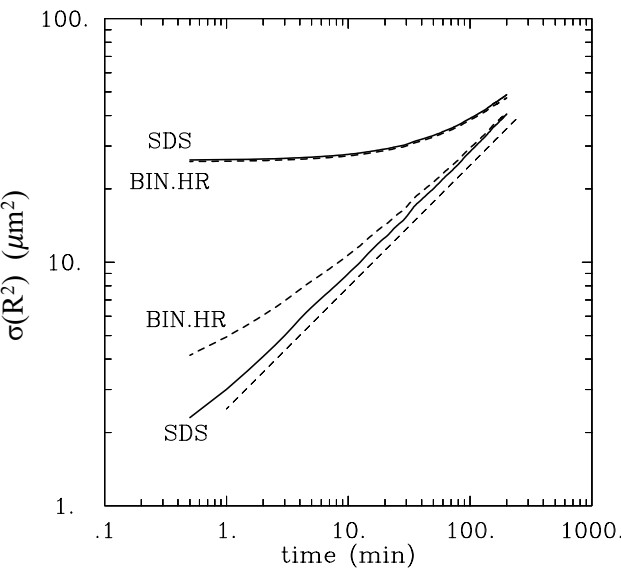

**Figure A1. Evolution of the radius squared standard deviation for extended superdroplet (solid lines) and bin (dashed lines) simulations with 130 cm⁻³ droplet concentration and low TKE. The lower lines are for initially-monodisperse simulations as in Figs. 6 (SDS) and 8 (BIN.HR) but extended from 20 to 200 minutes. The upper two lines are for simulations with truncated Gaussian initial droplet size distribution. Thin dashed line shows expected t$^{1/2}$ scaling.**

To understand the results of the truncated Gaussian simulations, we show in Fig A2 the droplet distributions at the onset of the simulations and after 10 minutes from the bin simulations. (Results from Lagrangian simulations are practically the same and thus they are not shown). When applying the linear vertical scale (left panels), the spectra look almost the same. However, panels with the logarithmic vertical scale show that the key change during the initial 10 minutes

of the simulations is an expansion of the spectra into tails that are barely visible on the left

panels. Note that formation of the tails insignificantly affects the spectral width which explains the evolutions shown in Fig. A1. The key point is that formation of the spectral tails beyond the truncated Gaussian initial distribution and transition to the increasing spectral width proceeds gradually and in virtually the same way in both Eulerian and Lagrangian simulations.


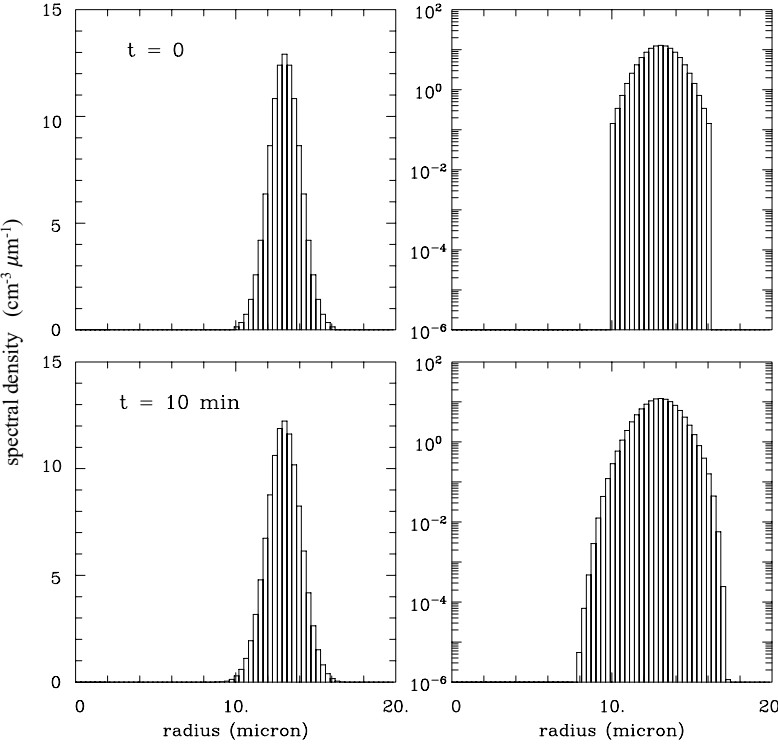

**Figure A2. Initial droplet spectra (upper row) and spectra at time of 10 minutes (lower row) for bin simulations. Left and right panels show spectra using linear and logarithmic vertical scale, respectively.**

## Appendix B. Summary of Eulerian and Lagrangian microphysics equations.

*Bin microphysics*

The bin scheme solves the equation for the spectral density function $\psi(r, x) = dN(x)/dr$, where $dN(x)$ is the concentration of droplets at spatial location $x$ and in the radius interval $(r, r + dr)$. Without droplet activation, sedimentation, and collisions, the continuity equation for $\psi$ is:

$$\frac{\partial \psi}{\partial t} + \text{div}\,(\boldsymbol{u}\psi) + \frac{\partial}{\partial r}\left(\frac{dr}{dt}\psi\right) = 0$$

where $\boldsymbol{u}$ is the fluid velocity and $dr/dt$ is the diffusional droplet growth rate (9). In the discrete system consisting of $\mathcal{N}$ bins of droplet sizes, the spectral density function for each bin $i$ with the radius $r^{(i)}$ is defined as $\psi^{(i)} = N(i)/\Delta r^{(i)}$, where $N(i)$ is the concentration of droplets in the bin $i$ and $\Delta r^{(i)}$ is the bin width. This transforms the continuous equation into a system of $\mathcal{N}$ coupled equations:

$$\frac{\partial \psi^{(i)}}{\partial t} + \text{div}\left(\boldsymbol{u}\psi^{(i)}\right) = \left(\frac{\partial \psi^{(i)}}{\partial t}\right)_{\text{cond}}$$

where the term on the right-hand side represents the condensational growth term, that is, the advective transport in radius space. It is calculated by combining all $\mathcal{N}$ bins at each grid volume as explained in the main text. The condensation rate $C_d(\boldsymbol{x})$ is calculated as

$$C_d = \frac{4\rho_w}{3\rho} \pi \sum_{i=1}^{\mathcal{N}} [r^{(i)}]^3 \left(\frac{\partial \psi^{(i)}}{\partial t}\right)_{\text{cond}} \Delta r^{(i)}$$

where $\rho_w = 10^3$ kg m$^{-3}$ is the liquid water density.

*Superdroplets*

The superdroplet scheme follows evolution in time and space of an ensemble of point particles that move with the flow and grow or evaporate in response to the grid volume supersaturation. The evolution of the $j$th superdroplet position $\boldsymbol{x}_j$ is calculated as

$$\frac{d\boldsymbol{x}_j}{dt} = \boldsymbol{u}(\boldsymbol{x}_j, t)$$

where $\boldsymbol{u}$ is the air flow velocity predicted by the dynamical model interpolated to the $j$ superdroplet position. Each superdroplet responds to the grid volume supersaturation and grows according to (9). The condensation rate is calculated as

$$C_d = \frac{d}{dt}\left(\sum_k \frac{4\rho_w}{3\rho} \pi r_k^3 N_k\right)$$

where summation is over all superdroplets within a given grid volume, and $r_k$ and $N_k$ are the $k$th superdroplet radius and multiplicity.

*Data availability*. Data supporting this study is available at
https://dashrepo.ucar.edu/dataset/144_grabow.html, (https://doi.org/10.5065/2ybq-nd09). Last
accessed 14 January 2021.

*Author contributions*. WWG developed the idea and ran the simulations. WWG and LT
performed data analysis and prepared the manuscript.

*Competing interests*. The authors declare that they have no conflict of interest.

*Acknowledgments.* Simulations presented in this manuscript were completed in summer 2020
and the manuscript was drafted in summer and fall 2020 during NCAR's compulsory work-
from-home period due to the COVID-19 pandemic. Initial flow data that were used to initiate the
simulations were provided by JPL's Dr. Marcin Kurowski. WWG acknowledges partial support
from the U.S. DOE ASR Grant DE-SC0016476. NCAR is sponsored by the National Science
Foundation.

*Financial support*. This research has been partially supported by the U.S. DOE ASR Grant DE-
SC0016476.

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
