# Peer review of "Cloud droplet diffusional growth in homogeneous isotropic turbulence: bin microphysics versus Lagrangian superdroplet simulations"

_Atmospheric Chemistry and Physics, 2020_

## Referee Comment (RC1) · Anonymous Referee #1 · 4 Dec 2020

General comments:

The present paper compares two different mathematical descriptions (Lagrangian and Eulerian frameworks) to address the cloud droplet diffusional growth in homogeneous isotropic turbulence. The manuscript shows interesting results with potential interest for Atmospheric Chemistry and Physics community. Nevertheless, the Reviewer has several comments/questions/suggestions that could make this paper even more useful for the community.

Specific comments:

-In order to have a more detailed (and useful) analysis, the Authors should report the final square radius distributions for all the Eulerian and Lagrangian cases. The theory shows that the pdf($R^2$) is Gaussian. Are they Gaussian or not in these cases? The referee expects a departure from the Gaussian distribution in high TKE SDS.26

-It is better to plot in the same graph the statistics from the bin and Lagrangian simulations for every single case so that we can directly compare one by one.

-The Reviewer is not convinced that the results of the bin simulations differ so much due to the difficulty to describe the monodisperse delta distribution as an initial condition. The Authors can easily run the Lagrangian simulations for some iterations, extract the radius distribution pdf and use as an initial condition for the bin simulations (together with the same flow, temperature and humidity fields). Then they should be able to reduce the errors induced by wrong initial conditions and maybe to analyse better the effects of the bin resolutions or the effects of numerical diffusion.

-Why the high TKE simulations are run just for few minutes compared with the low TKE cases? What happens for long times? Since the resolution and number of super-droplets/bins are the same, the computational efforts are precisely the same, so there are no problems to extend these cases in principle.

-The Authors should describe with more equations and details the Lagrangian and the bin approaches, so far, everything is referred to previous papers.

-For the bin simulations, a diffusion coefficient for the droplet distribution function should be given, how much is this value in the present results?

-Sedimentation and inertial terms are neglected in the Lagrangian simulations, what about in the Eulerian cases?

-Why does the term $r_0$ appear in equation (9)? It is needed to avoid some singularities when r is small? Is $r_0$ appearing also in the Lagrangian radius equation evolution? Why

has $r_0$ that specific value?

-The value of the dissipation in Lanotte et al. (2009) is $10^{-3}$ not $10^{-4}$ $m^2/s^3$

-It is better to introduce a new Table with the case description

---

## Referee Comment (RC2) · Anonymous Referee #2 · 9 Dec 2020

Comments to the manuscript with ID number "acp-2020-1106"

General comments:

The authors of this manuscript compared the Eulerian binning method and Lagrangian superdroplet approach in simulating the condensation process of cloud droplet driven by turbulence. They concluded that the Lagrangian superdroplet approach is able to represent fluctuations better, which is consistent with previous works as discussed in this manuscript. This detailed comparison between the two numerical method could

help contribute to a better understanding of modelling the condensation process.

However, in my opinion, the so called "ill-posedness of the initially monodisperse droplet size distribution for the bin microphysics." described in this manuscript could be avoided by testing different initial droplet-size distributions. I would recommend the publication of this manuscript after the authors carefully address this problem and other few comments listed below.

Specific comments:

L.24: turbulence integral time scale; L.145: Could the authors use mathematical symbols in equations all across the manuscript (e.g. Eq.1 and 2) to improve the readability of the manuscript? L.280: Kolmogorov slope. L.412: What is the equation to calculate C\_d? L.630: What is the difference between the two plots at the lower panels?

Technical corrections:

L.13: by applying L.96: based on L.102: point-by-point L.412: due to L.545: High L.575: ...show the expected ... L.585: macro L.613: being present. What is q\_c 40/N?

**ACPD**

---

## Author Comment (AC1) · 8 Jan 2021

**Responses to Reviewer 1 comments.**

Original comments in black, our responses and explanation of changes in red.

The present paper compares two different mathematical descriptions (Lagrangian and Eulerian frameworks) to address the cloud droplet diffusional growth in homogeneous isotropic turbulence. The manuscript shows interesting results with potential interest for Atmospheric Chemistry and Physics community. Nevertheless, the Reviewer has several comments/questions/suggestions that could make this paper even more useful for the community.

**Specific comments:**

-In order to have a more detailed (and useful) analysis, the Authors should report the final square radius distributions for all the Eulerian and Lagrangian cases. The theory shows that the pdf(R2) is Gaussian. Are they Gaussian or not in these cases? The referee expects a departure from the Gaussian distribution in high TKE SDS.26.

Motivated by this comment, we plotted radius squared  $(R^2)$  distributions for all simulations. These spectra are close to Gaussian, and some examples are included in the revised text (new section 5.3 included at the bottom of the document for the Reviewer's reference). We feel this addition significantly adds to the manuscript and we are thankful to the Reviewer for this suggestion.

-It is better to plot in the same graph the statistics from the bin and Lagrangian simulations for every single case so that we can directly compare one by one.

We appreciate this suggestion. However, such a change requires a significant modification of the manuscript structure. We prefer to avoid that. The way Lagrangian and Eulerian results are plotted allows a direct comparison without plotting the same droplet concentration cases on the same figure. Moreover, there are several lines for the bin results already and adding Lagrangian results would make such a plot very crowded.

However, per the Reviewer suggestion above, we added  $R^2$  spectra that are now discussed in a separate section, with Lagrangian and Eulerian results next to each other. We feel the addition section allows a good comparison between the results.

-The Reviewer is not convinced that the results of the bin simulations differ so much due to the difficulty to describe the monodisperse delta distribution as an initial condition. The Authors can easily run the Lagrangian simulations for some iterations, extract the radius distribution pdf and use as an initial condition for the bin simulations (together with the same flow, temperature and humidity fields). Then they should be able to reduce the errors induced by wrong initial conditions and maybe to analyze better the effects of the bin resolutions or the effects of numerical diffusion.

We agree with this comment. However, as documented in the new section 5.3, the bin spectra are well resolved after some time when appropriately high bin resolution is applied. This is because the initially monodispersed spectrum evolves in  $R^2$  Gaussian spectrum as the simulation progresses. We hope the added section 5.3 with  $R^2$  spectra addresses this point. We include the new section at the end of this document.

That said, and motivated by the Reviewer 2 comments, we ran additional simulations and drafted an appendix that can be added to the manuscript. At the moment, we feel this is not needed, but we will be happy to include such an appendix in the revised manuscript. Please see the draft appendix at the end of our responses to the Reviewer 2 comments.

-Why the high TKE simulations are run just for few minutes compared with the low TKE cases? What happens for long times? Since the resolution and number of superdroplets/bins are the same, the computational efforts are precisely the same, so there are no problems to extend these cases in principle.

The low and high TKE cases were run for the same nondimensional time (6 eddy turnover times) and their computational effort is exactly the same. The nondimensional time is shown at the bottom of original figures 4 and 7. We feel this is long enough for the goal of this manuscript. Please also see the discussion in the new section 5.3 discussing the  $R^2$  spectra.

-The Authors should describe with more equations and details the Lagrangian and the bin approaches, so far, everything is referred to previous papers.

Both Lagrangian and Eulerian approaches to cloud microphysics are fairly standard in the cloud physics literature. Thus, we feel including model equations would only add text that is not really needed. We prefer the style that we follow: discuss model details by words and refer to published papers where mathematical formulas are given.

However, as stated in our responses to the Reviewer 2 comments, we are open to include the equations if both Reviewers insist.

-For the bin simulations, a diffusion coefficient for the droplet distribution function should be given, how much is this value in the present results?

We are not sure what the diffusion coefficient the Reviewer has in mind. The two schemes apply the same droplet growth equations, so the molecular diffusion coefficient (implicit in the droplet growth equation) is the same for both schemes. For the numerics, the bin scheme does not use any diffusion coefficient per the ILES approach.

-Sedimentation and inertial terms are neglected in the Lagrangian simulations, what about in the Eulerian cases?

The Eulerian scheme excludes those as well. This is mentioned around line 405 of the original submission. This is appropriate for the spatial scales resolved by the simulations.

-Why does the term r0 appear in equation (9)? It is needed to avoid some singularities when r is small? Is r0 appearing also in the Lagrangian radius equation evolution? Why has r0 that specific value?

This is to include kinetic effects that play role at very small droplet radii. We added a reference to an old paper by Mordy (Tellus 1959) and a couple more recent papers that include such a formulation (by Clark and by Kogan). As stated in the original manuscript, the same droplet growth equation is used in both schemes (see the end of section 4).

-The value of the dissipation in Lanotte et al. (2009) is 10-3 not 10-4 m2/s3.

Corrected. Thank you.

-It is better to introduce a new Table with the case description.

We added a new table (Table 1) in section 4 with simulation details.

**The new section:**

**5.3** Comparison of radius squared distributions between Eulerian and Lagrangian simulations.**

This section compares radius squared ( $R^2$ ) distributions at the end of the simulations, that is, after 6 turnover times, for both the low and high TKE simulations. As shown in Lanotte et al. (2009) and Sardina et al. (2015), an initial monodisperse distribution should evolve into a Gaussian  $R^2$  spectrum because of the parabolic cloud droplet growth equation. Although the parabolic growth is only approximately valid because of the specific droplet growth equation (see Eq. 9), the Gaussian distribution is a good fit for simulation results discussed here as shown below.

Figure 9 shows the spectra for selected superdroplet simulations. The radius squared spectra are created by selecting  $R^2$  bin size and binning superdroplet radii for a given simulation into the assumed bin grid. The bin size for the SDS.650/SDS.26 simulations (lower/upper panels in Fig.9) is 1/10  $\mu$ m2. There are two panels for each simulation, one with the linear vertical scale and the spectrum shown as a histogram, and the second one with the logarithmic vertical scale and using star symbols to show the spectrum. In addition, the logarithmic plots show the Gaussian distributions obtained with the mean and standard deviation calculated from the spectra.

Figure 9. Results from simulations (upper panels) SDS.26 and (lower panels) SDS.650 superdroplet simulations. There are two panels for each simulation, the left one applying the linear vertical scale and the right one applying the logarithmic scale. The line in the logarithmic scale panels shows the Gaussian distribution with the mean and standard deviation calculated from the spectrum. Left/right pair in each row is for low/high TKE simulation.

For the SDS.650 simulations (lower panels in Fig. 9), the spectra at the end of low and high TKE simulations are practically the same. This agrees with the theoretical scaling and simulation results shown in Fig. 4 and 6. In contrast, results for SDS.26 differ drastically between the low and high TKE. The spectrum for the low TKE is wide, with some small droplets already evaporated because the spectrum is truncated at the low-radius end. Nevertheless, the Gaussian shape is still a good fit for the simulated spectrum. The high TKE SDS.26 spectrum is significantly narrower with small deviations from the Gaussian fit.

Figure 10. As Fig. 9, but for the bin (upper panels) BIN.26 and (lower panels) BIN.VHR.650 simulations.

Figure 10 shows the spectra for bin simulations similar to those in Fig. 9. Since bin simulations predict the spectra directly, the radius spectra are converted to  $R^2$  spectra and then plotted at their native resolution in the  $R^2$  space. This explains the change in the resolution along the horizontal axes evident in the upper panels. Overall, there are some similarities between Figs. 9 and 10. For instance, upper panels show spectra for the 26 cm-3 simulations with 0.5-µm bin width that are similar to those in superdroplet simulations. Spectra for 650 cm-3 simulations with 0.1-µm bin width (i.e., from the VHR set) are also similar between low and high TKE simulations, but their spectral widths are larger than in corresponding panels of Fig. 9. The impact of the bin resolution is further documented in Fig. 11 that shows results from the 650 cm-3 low TKE HR and SHR simulations, that is, with the bin width of 0.3 and 0.05 µm, respectively. Only the SHR simulation (i.e., the right panel in Fig. 11) resembles the spectra from the Lagrangian simulations shown in the lower panels of Fig. 9.

---

## Author Comment (AC2) · 8 Jan 2021

**Responses to Reviewer 2 comments.**

Original comments in black, our responses and explanation of changes in red.

The authors of this manuscript compared the Eulerian binning method and Lagrangian superdroplet approach in simulating the condensation process of cloud droplet driven by turbulence. They concluded that the Lagrangian superdroplet approach is able to represent fluctuations better, which is consistent with previous works as discussed in this manuscript. This detailed comparison between the two numerical method could help contribute to a better understanding of modelling the condensation process. However, in my opinion, the so called "ill-posedness of the initially monodisperse droplet size distribution for the bin microphysics." described in this manuscript could be avoided by testing different initial droplet-size distributions. I would recommend the publication of this manuscript after the authors carefully address this problem and other few comments listed below.

We appreciate the Reviewer's time and effort to read and provide comments on our manuscript. In the revision, we actually decided to tone down the discussion of the ill-posedness of the initial condition in the Eulerian scheme. This is because, as shown in the revised text, the distribution becomes well-resolved after some time, and bin microphysics matches nicely superdroplet solutions at the end of the simulations once the bin resolution is sufficiently high. We start with a monodisperse distribution because previous studies of Lanotte et al., Sardina et al. and Thomas et al. considered exactly the same monodisperse initial distribution.

Our initial thought was to follow the reviewer's advice and complete some simulations with initial Gaussian-like distribution. We thought about including those results as an appendix. However, after analyzing the spectra (per Reviewer's 1 suggestion), we decided that this is not needed. We include a discussion of those results at the end of this response for the Reviewers' reference. We are open to the suggestion of either including or excluding those results in the revised manuscript.

Specific comments:

L.24: turbulence integral time scale; Added "integral" in the sentence in question.

L.145: Could the authors use mathematical symbols in equations all across the manuscript (e.g. Eq.1 and 2) to improve the readability of the manuscript?

This comment echoes one of the comments of the Reviewer 1. However, we strongly feel that explaining in words details of the two microphysical schemes (with references to previous papers that present the equations) rather than through specific equations should be sufficient for the reader. We will be happy to include the equations if the reviewer insists.

L.280: Kolmogorov slope. Added "Kolmogorov" in the sentence in question.

L.412: What is the equation to calculate C\_d? We slightly modified the description, but feel - as explained above - that using words is sufficient.

L.630: What is the difference between the two plots at the lower panels? This is explained in the text that we slightly modified. The point is that because of statistical fluctuations of the number of superdroplets per grid box N (when compared to the expected value of 40), the cloud water strongly fluctuates in the Lagrangian scheme. The magnitude of the fluctuation can be reduced by rescaling the cloud water calculated from the superdroplets present within a grid box by a factor of 40/N. The lower panels show the results before (the left panel) and after (the right panel) such a rescaling.

**Technical corrections:**

L.13: by applying L.96: based on L.102: point-by-point L.412: due to L.545: High L.575: . . .show the expected . . . L.585: macro L.613: being present. What is  $q_c 40/N$ ? The changes were included in the text as suggested. The last point was addressed by explaining in the revised text that 40 is the mean number of superdroplets expected in each grid volume (see our response to L.630 comment above).

**Possible Appendix.**

This appendix describes results from additional Eulerian bin and Lagrangian superdroplet low TKE simulations for the case of the total droplet concentration of 130 cm-3 with the initial either monodisperse droplet spectrum (i.e., as in the main text) or a wide spectrum that can be well represented by the bin microphysics. For the latter, a truncated Gaussian spectrum is used with the width of 1  $\mu$ m and truncated to zero for droplet radius outside the 10 to 16  $\mu$ m range (i.e., three standard deviations). For the bin microphysics, the bin setup is as in the BIN.HR.130 (i.e., high-resolution) simulation, that is, with the bin width of 0.3  $\mu$ m. With a single bin centered at 13  $\mu$ m, there are 21 nonzero bins for the truncated Gaussian distribution in the 40-bin Eulerian setup. To simplify the super-droplet setup, we use 42 superdroplets per grid volume (rather than 40 as in main-text simulations), repeating twice the 21 superdroplet radii and multiplicities to exactly match the non-zero 21 bins in the initial bin setup. Motivated by the initial results, we extend ten times the simulation length, that is, up to 200 min, over 60 eddy turnover times. The results are documented in Figs. A1 and A2.

Figure A1 shows evolution of the radius squared standard deviation in the format of Figs. 6 and 8 of the main text for simulations with monodisperse and with truncated Gaussian initial conditions. Extending ten times the length of simulations from the main text shows that the Lagrangian simulation continues on the expected  $t^{1/2}$  scaling, and the bin simulation continues to approach that scaling. The truncated Gaussian initial droplet distribution simulations are close to each other, and they start to approach the expected scaling only after the initial 10 min.

Figure A1. Evolution of the radius squared standard deviation for extended superdroplet (solid lines) and bin (dashed lines) simulations with 130 cm-3 droplet concentration and low TKE. The lower lines are for initially-monodisperse simulations as in Figs. 6 (SDS) and 8 (BIN.HR) but extended from 20 to 200 minutes. The upper two lines are for simulations with truncated Gaussian initial droplet size distribution. Thin dashed line shows expected t1/2 scaling.

Figure A2. Initial droplet spectra (upper row) and spectra at time of 10 minutes (lower row) for bin simulations. Left and right panels show spectra using linear and logarithmic vertical scale, respectively.

To understand the results of the truncated Gaussian simulations, we show in Fig A2 the droplet distributions at the onset of the simulations and after 10 minutes from the bin simulations. (Results from Lagrangian simulations are practically the same and thus they are not shown). When applying the linear vertical scale (left panels), the spectra look the same. However, panels with the logarithmic vertical scale show that the key change during the initial 10 minutes of the simulations is an expansion of the spectra into tails that are barely visible on the left panels. Note that formation of the tails insignificantly affects the spectral width which explains the evolutions

shown in Fig. A1. The key point is that formation of the spectral tails beyond the truncated Gaussian initial distribution and transition to the increasing spectral width proceeds gradually and in virtually the same way in both Eulerian and Lagrangian simulations.

---

## Author Response (AR2)

**Responses to Reviewers' final comments.**

Comments from Reviewer 1:

The Authors have sufficiently addressed my previous concerns and I feel that the Manuscript has been improved and I recommend publication of this work. Regarding the description of the two numerical frameworks, the Authors can create a Appendix/Supplemental Information section with all the details without the need to change the structure and readability of the main text.

Comments from Rev 2:

Since cloud droplets are rarely mono-dispersed in atmospheric clouds, adding the appendix proposed by the authors could improve and complete the key point of this manuscript. I would suggest the authors to write down the equation for C_d such that readers from other communities can also understand it without looking into previous studies. I would recommend the publication of this manuscript after the authors consider these two points.

In response to the above comments and suggestion by the Editor, we included two appendices into the final manuscript. One was attached to our responses and included discussion of results with well-resolved initial conditions. The second appendix includes microphysics schemes equations, including the condensation rate formulation.